# Cyclic Dipeptides: The Biological and Structural Landscape with Special Focus on the Anti-Cancer Proline-Based Scaffold

**DOI:** 10.3390/biom11101515

**Published:** 2021-10-14

**Authors:** Joanna Bojarska, Adam Mieczkowski, Zyta M. Ziora, Mariusz Skwarczynski, Istvan Toth, Ahmed O. Shalash, Keykavous Parang, Shaima A. El-Mowafi, Eman H. M. Mohammed, Sherif Elnagdy, Maha AlKhazindar, Wojciech M. Wolf

**Affiliations:** 1Faculty of Chemistry, Institute of General & Inorganic Chemistry, Technical University of Lodz, 90-924 Lodz, Poland; wojciech.wolf@p.lodz.pl; 2Institute of Biochemistry and Biophysics, Polish Academy of Sciences, Pawinskiego 5a, 02-106 Warsaw, Poland; amiecz@ibb.waw.pl; 3Institute for Molecular Bioscience, The University of Queensland, St. Lucia, QLD 4072, Australia; z.ziora@imb.uq.edu.au (Z.M.Z.); i.toth@uq.edu.au (I.T.); 4School of Chemistry and Molecular Biosciences, The University of Queensland, St. Lucia, QLD 4072, Australia; m.skwarczynski@uq.edu.au (M.S.); a.shalash@uq.net.au (A.O.S.); 5School of Pharmacy, The University of Queensland, Woolloongabba, QLD 4102, Australia; 6Center for Targeted Drug Delivery, Department of Biomedical and Pharmaceutical Sciences, Harry and Diane Rinker Health Science Campus, School of Pharmacy, Chapman University, Irvine, CA 92618, USA; parang@chapman.edu (K.P.); elmowafi@chapman.edu (S.A.E.-M.); emohammed@chapman.edu (E.H.M.M.); 7Botany Department, Faculty of Science, Cairo University, Giza 12613, Egypt; sh.elnagdy@gmail.com (S.E.); malkhazi@aucegypt.edu (M.A.)

**Keywords:** cyclic dipeptides, diketopiperazines, proline-based DKPs, drug discovery, privileged scaffold, supramolecular structuring

## Abstract

Cyclic dipeptides, also know as diketopiperazines (DKP), the simplest cyclic forms of peptides widespread in nature, are unsurpassed in their structural and bio-functional diversity. DKPs, especially those containing proline, due to their unique features such as, inter alia, extra-rigid conformation, high resistance to enzyme degradation, increased cell permeability, and expandable ability to bind a diverse of targets with better affinity, have emerged in the last years as biologically pre-validated platforms for the drug discovery. Recent advances have revealed their enormous potential in the development of next-generation theranostics, smart delivery systems, and biomaterials. Here, we present an updated review on the biological and structural profile of these appealing biomolecules, with a particular emphasis on those with anticancer properties, since cancers are the main cause of death all over the world. Additionally, we provide a consideration on supramolecular structuring and synthons, based on the proline-based DKP privileged scaffold, for inspiration in the design of compound libraries in search of ideal ligands, innovative self-assembled nanomaterials, and bio-functional architectures.

## 1. Introduction

Cyclic dipeptides, also known as cyclo-dipeptides, diketopiperazines (DKPs), piperazinediones, dioxopiperazines, or dipeptide anhydrides, are the simplest, naturally occurring cyclic forms of peptides, commonly biosynthesized by a large variety of living organisms [1,2,3,4] and conserved in bacteria to humans [5,6]. They were first discovered in 1880 and later studied by E. Fischer [7]. Once believed to be only protein artifacts or degradation products, and therefore neglected, DKPs are now considered essential metabolic intermediates, and an interesting platform for therapeutic exploration [8]. DKPs possess all advantages of cyclic peptides. DKPs are an unsurpassed class of bio-molecules in their structural and bio-functional diversity. Moreover, the ‘*biosynthetic hooks*’ are a useful strategy for the identification of the genes modifying the DKP ring to expand the chemical space of cyclic dipeptides [9,10]. Moreover, modified DKPs have recently emerged as an important pharmacophore in a number of theranostic settings. Notably, proline motif introduces additional conformational and bio-functional value into the DKP-derived structures. The attractive features, such as extra rigidity, structural stability, and consequently, greater resistance to degradation by enzymes, higher bioactivity, specificity, selectivity, and efficacy, increased cell permeability, or binding affinity to specific targets, inherent biocompatibility, structural importance to biological systems [11,12] make proline-based cyclic dipeptides a promising alternative to currently used small molecule and macromolecule pharmaceuticals.

The mechanism of proline-based DKPs formation is described elsewhere [13].

DKPs are ‘diamonds in the rough’, offering endless possibilities in future innovative therapies [14]. Therefore, since the earliest report on DKP in 1924 [15], an increasing degree of attention to DKPs has been observed, and numerous scientific findings revealing their broad-spectrum biological activities have been reported in recent years. In particular, proline-based DKPs have diverse properties depending on structure and application, such as anticancer, antioxidant, neuroprotective, antiviral, antibacterial, anti-inflammatory, antihyperglycemic, antiarrhythmic, immunomodulatory, antiparasitic, anthelmintic, insecticidal, antifouling, vasorelaxant, and metabolic regulatory activity [16,17,18,19,20,21,22]. They have the potential to be antibiotics of the future. Moreover, they act as antagonists of human oxytocin receptors [23], inhibitors of platelet aggregation [24], calpain inhibitors against stroke [25]. DKPs have relevance in the prevention of cell division [26], cell–cell signaling, or quorum sensing [27]. They are useful in smart delivery systems of drugs that have low permeability to cross the blood-brain barrier [19]. As a curiosity, DKPs are a hot topic in ecological chemistry [28]. Furthermore, DKP-containing compounds are used as catalysts or chiral auxiliaries in synthetic organic chemistry, in the formation of alkaloids [29,30,31]. They are an excellent model in theoretical studies on the constrained structural scaffold with a relevant pharmacophore [17,32,33]. In the future, cyclic-dipeptide-based compounds will find a wider use in various fields as perfect tools for probing specific proteins or metabolites in vivo, or as building blocks for macromolecules.

Here, we present a comprehensive overview of the recent progress on bio-landscape and structural diversity of compounds containing proline-based DKP motif, which are exploited as privileged peptidomimetic scaffolds for future innovative drug discovery, smart delivery systems, and modern bio-control agents [34,35,36]. We pay special attention to anticancer proline-based DKPs since cancers are main cause of death all over the world, with nearly 10 million deaths in 2020 according to the WHO [37]. Therefore, novel, effective, and safe therapeutics are extremely needed. In this context, the supramolecular structuring and the role of versatile synthons for inspiration in the design of compound libraries in search of ideal ligands with unique proline-DKP motif are also discussed.

## 2. Occurence and Biosynthesis of DKPs

DKP skeleton is observed in micro-species, bacteria, such as *Bacillus subtilis*, *Streptomyces, Pseudomonas aeruginosa*, or *Lactobacillus plantarum* [38,39,40], marine, and terrestrial fungi [41] as *Aspergillus flavus* or *Alternaria alternata*, and *Penicillium*, respectively [42,43], marine sponges such as *Dysidea herbacea*, and *fragilis* [44], or proteobacteria *Alcaligenes faecalis*, algae, lichens, gorgonians, tunicates, plants, or animals venoms. Remarkably, DKPs were found in human central nervous system, gastrointestinal tract, or blood [45]. They occur in food and beverages, such as pu-erh tea, cocoa, dried bonito, roasted coffee, sake, beer, cheese, casein, chicken extract, or stewed beef, giving a special metallic bitter taste [13,18,31,32,46,47,48,49], but also in culture broths fermented with lactic acid bacteria. Products containing both l- and d-proline-based DKPs are common in nature, and their structural and biological complexity is highly impressive [50]. As a curiosity, 90% of DKPs in foods contain proline [28]. DKPs provide an eco-friendly approach to food and feed preservation [51]. On the other hand, DKP framework is present in drugs, e.g., in aminopenicillin, amoxicillin, ACE inhibitors [52,53] as by-products of spontaneous intramolecular cyclization of the dipeptidyl moiety in active peptide-based substances. Degradation via DKPs formation is observed during long-term storage. Moreover, DKPs can appear as a result of chemical peptide synthesis, or hydrolysis of functional peptides and proteins [54,55]. The cyclization is facilitated when a proline is present at the second position from the N-terminus (i.e., penultimate proline) [55].

From the biosynthetic point of view, naturally produced DKPs are known to be effective and biodegradable, however their production yield is low [56,57]. Laboratory trials for DKPs-microbial induction have faced some constraints. Although producing DKPs from microorganisms via an expression system is feasible; the optimization is long and not a straightforward process [58]. The biosynthesis of DKPs relies mainly on two enzymes, non-ribosomal peptide synthetases (NRPs) and tRNA-dependent cyclodipeptide synthases (CDPs) [59]. Both enzymes are part of a biosynthetic gene cluster that targets DKPs scaffold modifications necessary for the stability of the produced DKPs [60]. Metagenomics and next generation sequencing enhanced the biosynthetic gene clusters encoding DKP tailoring enzymes [61]. As reported, the microbial genes responsible for a specific secondary metabolite were found to be close to other genes in the dedicated biosynthetic gene clusters [62]. Since the genes responsible for DKPs biosynthesis are clustered on the microbial chromosome, therefore, the encoding of the biosynthetic genes depends on the discovery of a single gene in the pathway. There are approximately 700 known CDPs-encoding genes clustered with the predicted tailoring genes [9]. Recently, heterologous expression, zinc finger nuclease (ZFN) and transcription activator-like effector nuclease (TALEN) have been used to edit target genes for secondary metabolite induction in microorganisms [63]. However, these approaches found some limitations in their efficiency and productivity level [64,65,66]. Thus, clustered regularly interspersed short palindromic repeats (CRISPR)/associated protein (Cas) system has been recently used as a new approach for the biosynthesis of secondary metabolites and for activation of silent biosynthetic gene clusters [67]. CRISPR/Cas system has outmatched other techniques due to its possible multi-gene editing and high efficiency. Precisely, type II CRISPR/Cas system has been successfully applied for the biosynthesis of secondary metabolites [67]. Previous studies reported the use of CRISPR/Cas9 system in the filamentous fungus *Trichoderma reesei*, achieving the homologous recombination of > 93% efficiencies [68]. Similarly, Nodvig et al. [69] obtained a genome-edited phenotype by targeting the *yA* gene in the model fungus *Aspergillus nidulans*. Thus, CRISPR/Cas system could be a potential mechanism for the efficient biosynthesis of DKPs.

## 3. DKP Scaffold

The concept of scaffold is useful in medicinal chemistry and drug design to generate, characterize, and compare cores of bio-active substances and their analogs [70]. The scaffold is the main fragment of compound (e.g., ring system) after removal of substituents (*R*-groups) [71,72]. The privileged molecular scaffold is defined as a core structure, which forms structurally diverse bio-molecules via introducing different functional groups [73,74]. DKPs can play the role of a privileged, multi-functionalized scaffold for the design and development of advanced therapeutic agents, drug delivery systems, biomaterials, or bio-imaging to mitigate numerous disease conditions, also as for the synthesis of complex natural products [74] because of their specific conformational and physico-chemical attributes. DKPs are heterocyclic compounds consisting of two amino acid residues linked to a central six-membered lactam ring core with (or without) various substituents, providing the control of the substituent’s stereochemistry at up to four positions, chiral nature, three-dimensionality, and consequently leading to the promotion of the intermolecular H-bonding interactions with bio-target sites via the corresponding sites of donors and acceptors [17]. Thus, the rigid DKP core allows either a constrained or flexible behavior of amino acids, mimicking preferential peptide conformation [59]. It makes DKP molecules perfect to predicting properties of larger peptides with multiple H-bond acceptors, and donor functionality, and multiple sites for the structural elaboration of diverse functional groups. These characteristics do not only enable them to bind with high affinity to a large variety of receptors and enzymes [32], showing multifarious biological activities, but they also allow a more predictable receptor interaction and the development of the drug-like physicochemical properties that are required for the multi-objective optimization process of transforming lead to a drug product. The general structure for DKP cores can be seen in Figure 1. It should be highlighted that even though 2,5-DKPs are the most popular [32,75], other two regioisomers, such as 2,3-DKPs, and 2,6-DKPs, are also possible as important pharmacophores [76]. All isomers can be found in natural sources in the course of biochemical synthesis. Interestingly, the first crystal structure of DKP, 2,5-DKP, was reported in 1938 [19,77]. More specifically, 2,5-DKPs are common, naturally occurring peptide derivatives (and are frequently generated as unwanted by-products in the synthesis of oligopeptides). The 2,5-DKP core is present in the structure of known drugs, such as Tadalafil, phosphodiesterase-5 inhibitor for the treatment of pulmonary arterial hypertension and erectile dysfunction [78,79,80], Retosiban, an oxytocin antagonist for preterm labor [81], Epelsiban, an oxytocin antagonist in premature ejaculation in men [32], Aplaviroc against HIV [82], in the vascular disruption, and tubulin-depolymerizing Plinabulin, on the basis of marine fungal Halimide, a potential therapeutical drug in lung cancer [83,84], and other anticancer natural agents as Ambewelamide, Phenylahistin, Dehydrophenylahistin [85], Verticillin A [86], antiviral and immunosuppressive Sirodesmin, a type of phytotoxin, anti-inflammatory agents, e.g., FR106969 [59], antibacterial Bicyclomycin, Brevianamide S, Avrainvillamide [87] or Albonoursin, antifungal Maremycin, mycotoxins such as Roquefortine C [22] or Gliotoxin, which is also a potent inducer of apoptotic, and necrotic cell death [88,89].

Thaxtomin A [90], phytotoxin and insecticidal okaramine, and so on. Furthermore, 2,5-DKPs are present in food, e.g., in fermented olives and beverages. Thus, they have the potential to be used in the development of new functional foods [91]. Diverse 2,5-DKPs have been discovered from marine habitats (sponges, and microorganisms) in recent years [14]. These DKPs have cytotoxic (~36%), antimicrobial (~19%), antiviral (~13%), antioxidant (10%), enzyme inhibition (~5%), and other activities (18%) [14].

2,3-DKPs can be found in natural products, e.g., antibiotics, Piperacillin, or Cefoperazone [92]. Like 2,5 DKPs, they have been used in medicinal chemistry, for example, against diseases wherein platelet agglutination participates [93]. The first synthesis of 2,3-DKP was reported by Goulding and Pollard in 1948 [94].

2,6-DKPs have been investigated as antiproliferative agents through the inhibition of DNA topoisomerase [32,92,95]. They also have other activities, such as anticonvulsant [96] or trypanocidal [96,97,98,99].

Proline-based DKPs have a special characteristic [12]. Proline is a unique amino acid due to its specific structure. The DKP nucleus is fused to the pyrrolidine ring, resulting in eminent bio-properties, mentioned earlier. Both l- and d-proline, and to a lesser extent hydroxyproline-based DKPs, exhibit bio-activity. Moreover, proline cis-trans isomerization play a role inter alia in controlling auto-inhibition of signaling proteins [12]. In nature, proline-based DKPs (e.g., cyclo(l-Pro-l-Pro), cyclo(l-His-l-Pro), cyclo(l-Phe-l-Pro), cyclo(l-Tyr-l-Pro), cyclo(l-Leu-l-Pro), and cyclo(l-Val-l-Pro)) are common, which is translated into the structural complexity and impressive bio-activities of this important class of DKPs [51,91,100,101,102,103], which are thoroughly described in the next sub-section. Interestingly, cyclo(Pro-Pro) is as an archaic precursor in the early evolution of life process [5,38]. Notably, many biologically important cyclic peptide sequences, and natural products contain multiple proline residues. The ‘proline-rich cyclic structures’ have started gaining the attention of the pharmaceutical industry, but their true potential is still very much unknown. Analogs of proline in DKPs should not be overlooked. As an example, silaproline exhibits similar conformational properties, but confers higher lipophilicity and improved resistance to biodegradation [104].

## 4. Bio-Landscape and Structural Profile of Proline-Based DKPs

### 4.1. Anticancer Activity

#### 4.1.1. Bicyclic Proline-Based DKP with Simple Side Chains

Bi- and policyclic diketopiperazines containing a proline fragment within the structure are of a great interest to medical chemists as potential cytotoxic and antineoplastic agents. However, in most cases the cytotoxic effect of simple proline-containing bicyclic DKPs is quite limited. Ye [105] reported that cyclo (l-Phe-l-Hyp) (**1**), isolated from mangrove *Streptomyces* sp. Q24, inhibits the proliferation of human glioma U87-MG and U251 cells in at IC_50_ = 5.8 and 18.6 [μM], respectively (Table 1, entry 1). Cyclo (l-Phe-l-Hyp) (**1**), when tested on the adenocarcinoma HCT-116, the ovarian carcinoma OVCAR-8, and the glioblastoma SF-295 cell lines, did not exhibit any significant cytotoxic effect up to the concentration of 25 μg/mL (Table 1, entry 1) [106]. Cyclo (l-Leu-l-Hyp) (**2**) exhibits cytotoxic effect on U87-MG and U251 cells lines at IC_50_ = 14.5 and 29.4 [μM], respectively, while its close structural analog cyclo (l-Leu-l-Pro) (**3**) at IC_50_ = 1.3 and 19.8 [μM], respectively (Table 1, entries 2,3) [105]. Lin [107] reported that cyclo (l-Leu-l-Pro) (**3**), isolated from *Streptomyces xiamenensis* MCCC 1A01570, evaluated for cytotoxicity against three cancer cell lines of ECA-109 (esophageal carcinoma), HeLa-S3 (cervix carcinoma) and PANC-1 (pancreatic carcinoma) exhibited moderate inhibition effect at 20 μM varying from 14% (PANC-1) to 55% (ECA-109) (Table 1, entry 3). Shaala reported [108] that cyclo (l-Leu-l-Pro) (**3**), isolated from tunicate-derived actinomycete *Streptomyces* sp. moderately inhibits the proliferation of HCT-116, HepG2 (hepatocellular carcinoma) and MCF-7 (breast cancer) cell lines, with values of 16, ≥50, and 30 μg/mL, respectively (Table 1, entry 3). Its diastereoisomer, cyclo (d-Leu-l-Pro) (**4**), evaluated for cytotoxicity against ECA-109, HeLa-S3, and PANC-1 exhibited moderate inhibition effect at 20 μM varying from 44% (ECA-109) to 55% (PANC-1) [107] (Table 1, entry 4). The cytotoxic effect of cyclo (l-IIe-l-Pro) (**5**) was reported in two articles [107,108]. The compound **5** showed limited cell growth at 20 μM when tested on ECA-109, HeLa-S3 and PANC-1 cell lines, from 45% (HeLa-S3) to 56% (PANC-1) (Table 1, entry 5) [107], while inhibited the proliferation of HCT-116, HepG2 and MCF-7 cell lines, with values of 22, ≥50, and 27 μg/mL, respectively (Table 1, entry 5) [108]. Its hydroxylated analog, cyclo (l-IIe-l-Hyp) (**6**), when tested on ECA-109, HeLa-S3 and PANC-1 cell lines, inhibited the cell growth at 20 μM from 42% (PANC-1) to 54% (ECA-109) (Table 1, entry 6) [107]. Cyclo (4-*S*-hydroxy-d-Pro-d-Ile) (**7**) was isolated from Australian marine sponge *Stelletta* sp. and its cytotoxicity was tested on human tumour cell lines H460 (lung carcinoma), SF-268 (glioblastoma), MCF-7, HT-29 (colon adenocarcinoma) and a normal mammalian cell line CHO-K1, derived from hamster ovary. Compound **7** exhibited weak cytotoxic effect on all tested cell line with GI50 (μM) values varied from 204 (MCF-7) to > 295 (SF-268, CHO-K1) (Table 1, entry 7) [109]. Cyclo (l-Phe-l-Pro) (**8**) exhibited marked cytotoxicity, when tested on HCT-116, OVCAR-8 and SF-295, with IC_50_ μg/mL values of 21.4, 18.3, and 16.0, respectively [106]. The cytotoxicity observed effect of **8** was stronger than in the case of its hydroxylated derivarive **1** (Table 1, entries 1 and 8) [106]. The compound **8** also decreased the cell growth at 20 μM when tested on ECA-109, HeLa-S3, and PANC-1 cell lines, from 36% (HeLa-S3) to 50% (PANC-1) (Table 1, entry 8) [107]. Two stereoisomers of **8**: cyclo (l-Phe-d-Pro) (**9**) and cyclo (d-Phe-d-Pro) (**10**), as well as their 3-hydroxy analog, Penicicillatide B (**11**) (Table 1, entries 9–11), were isolated from the marine-derived fungus, *Penicillium* sp. and tested on the cytotoxic effect on three cancer cell lines: HCT-116, HepG2 and MCF7 [110]. Of the three cell lines tested, HCT116 proved to be the most sensitive to compounds **9–11**, with the IC_50_ [μM] values varied from 23.0 for Penicillatide B (**11**) to 94.0 for cyclo (d-Phe-d-Pro) (**10**). Cyclo (d-Phe-d-Pro) (**10**), a diastereoisomer of cyclo (l-Phe-d-Pro) (**9**), derived from d-Pro instead of l-Pro, exhibited about three times weaker effect then on HCT116 then **9** (IC_50_ = 94.0 vs. 38.9 μM), which proves that the configuration of the constituent amino acids may have a significant influence on the cytotoxic effect of the tested compound. Cyclo (l-Phe-2-OH-d-Pro) (**12**), a hydroxylated analog of cyclo (l-Phe-d-Pro) (**9**) was tested on three cancer cell lines: HCT-116, HepG2 and MCF7 (Table 1, entry 12) [108] and inhibited the proliferation with IC_50_ values of 30, ≥50, and 30 μg/mL, respectively. Wang reported that cyclo (l-Val-l-Pro) (**13**) (Table 1, entry 13), could inhibit HeLa cells with an inhibition rate of 33.3% at 100 μg/mL [111], while its hydroxylated analog Bacillusamide B (**14**), (Table 1, entry 14), inhibited the proliferation of HCT-116, HepG2 and MCF7 with IC_50_ values 25, ≥50, and 27 μg/mL, respectively [108]. Brevianamide F, cyclo (l-Trp-l-Pro) (**15**) exhibited marked cytotoxic effect on OVCAR-8 cell line (IC_50_ = 11.9 [μg/mL]) (Table 1, entry 15) [106], while its hydroxylated analog cyclo (l-Trp-l-Hyp) (**16**) showed moderate cytotoxic activity with IC_50_ = 64.34 [μM]) on HL-60 (acute promyelocytic leukemia) cell line (Table 1, entry 16) [112]. Cyclo (d-Leu-2-OH-Pro) (**17**) showed a rather weak cytotoxic effect with on HL-60 with IC_50_ = 98.49 [μM]) (Table 1, entry 16) [112], while Penicimutide (**18**) could inhibit HeLa cells with inhibition rate of 39.4% at 100 μg/mL (Table 1, entry 16) [111]. The simplest tricyclic proline-based DKP consists of two proline subunits, cyclo (l-Pro-l-Pro) (**19**) was evaluated for cytotoxicity against ECA-109, HeLa-S3, and PANC-1 (pancreatic carcinoma) exhibited moderate inhibition effect at 20 μM varying from 20% (ECA-109) to 40% (HeLa-S3) (Table 1, entry 19) [107]. Finally, Vázquez-Riviera reported [113] that the mixture of cyclo (l-Tyr-l-Pro) (**20**), cyclo (l-Val-l-Pro) (**13**), and cyclo(l-Phe-l-Pro) (**8**), isolated from *Pseudomonas aeruginosa* PAO1, initiated the cell death in HeLa and Caco-2 (colorectal adenocarcinoma) cell cultures with IC_50_ values of 0.53 and 0.66 mg/mL, respectively.

#### 4.1.2. Bicyclic Proline-Based DKP Modified with Indole-Based Side Chains

A number of bicyclic proline-based DKPs, bearing modified indole groups in the side chains were obtained from marine organisms. Tryprostatin A (**21**) and Tryptostatin B (**22**), isolated from marine fungal strain of *Aspergillus fumigatus* BM939, exhibited moderate inhibition effect when tested on H520 (squamous cell carcinoma), MCF7 and PC-3 (prostate adenocarcinoma) (Table 2, entries 1,2) [114]. The chemical modifications of Tryprostatins structures led to the discovery of diastereoisomer of Tryptostatin B (**22**), called ds2-TryB (**23**), possessing d-proline instead of l-proline moiety within its structure (Table 2, entry 3) [114,115]. Ds2-TryB (**23**) exhibited a very potent inhibitory effect on breast cancer resistance protein (BCRP), which was accompanied by a strong cytotoxic effect observed on the panel of 19 cancer cell lines, derived from both solid and blood tumors [116], while Tryprostatin A (**21**) and Tryptostatin B (**22**) exhibited an only moderate cytotoxic effect on H520, MCF7, and PC-3 cell lines at concentrations of 100 μM. The percent cell survival observed for ds2-TryB (**20**) at 100 μM varied from 0% (MCF7) to 0.2% (PC-3), and growth inhibition (GI_50_) in μM was established as 11.9 (H520), 17.0 (MCF7), and 12.3 (PC-3) (Table 2, entry 3) [115]. Piscarinine A (**24**) and Piscarinine B (**25**), isolated from the fungal strain of *Penicillium piscarium* VKM F-691, possess tri- or tetracyclic indole-based heterocycle in the side chain of DKP structure, as well as unsaturated, double bond in the proline ring (Table 2, entries 4,5) [116]. Initial results suggested that Piscarinine A (**24**) and Piscarinine B (**25**) exhibited a moderate cytotoxic effect on L929 (murine fibroblasts) and HeLa cell lines with IC_50_ values larger than 50 mg/mL [116]. Further research revealed that out of 36 tumor cell lines tested [117], LNCAP (prostate carcinoma) cell line seems to be the most susceptible to compounds **24** and **25**, with IC_50_ of 2.195 and 1.914 μg/mL, respectively (Table 2, entries 4,5). Notoamide C (**26**), prenylated indole alkaloids isolated from a marine-derived fungus, *Aspergillus* sp., showed moderate cytotoxicity against HeLa and L1210 (murine lymphocytic leukemia) cells with IC_50_ values of 50 and 22 μg/mL, respectively (Table 2, entry 6) [118]. Notoamide M (**27**) and its ethyl ether—17-*O*-ethylnotoamide M (**28**), isolated from co-culture of marine-derived fungi *Aspergillus sulphureus* and *Isaria feline*, significantly decreased colony formation of 22Rv1 (prostate carcinoma) cell line at concentrations of 10 μM by 55 and 25%, respectively (Table 2, entries 7–8) [119]. Finally, Brevianamide W (**29**), Brevianamide Q (**30**), Brevianamide R (**31**), Brevianamide K (**32**), Brevianamide E (**33**) were isolated from deep sea derived fungus *Aspergillus versicolor* CXCTD-06-6a and their cytotoxic effect was tested on P388 (murine leukemia), BEL-7402 (hepatocellular carcinoma) and MOLT-41 (acute lymphoblastic leukemia), but none of them showed cytotoxicity against the tested cell lines (Table 2, entries 9–13) [120].

#### 4.1.3. Tetracyclic Proline-Based DKP

Drimentidine G (**34**), tetracyclic, proline-based DKP possessed isopreonoid group, was isolated from *Streptomyces* sp. CHQ-64 [121] and tested for its cytotoxic effect on five cell lines: HCT-8 (ileocecal/colorectal adenocarcinoma), Bel-7402, BGC-823 (human papillomavirus-related endocervical adenocarcinoma), A549 (lung carcinoma), A2780 (ovarian carcinoma). Compound **34** exhibited promising cytotoxic effect with IC_50_ values of 2.81 ± 0.09, 1.38 ± 0.27, >10, 1.01 ± 0.04 and 2.54 ± 0.18 μM, respectively (Table 3, entry 1). In contrary to **34**, its lactam-methylated analogs, Drimentidine F (**35**) and Drimentidine H (**36**) did not exhibit promising activity up to 10 μM (Table 3, entries 2,3) [121,122].

Yu reported the isolation of tetracyclic okaramine C (**37**) and okaramine G (**38**) from coprophilous fungus *Aphanoascus fulvescens* and tested them for cytotoxic effect on L5178Y (mouse lymphoma) cell line (Table 3, entries 4,5) [123]. Compounds **37** and **38** showed significant cytotoxicity with IC_50_ values of 14.7 and 12.8 μM, respectively. Cai described novel Okaramines S-U, isolated from *Aspergillus taichungensis* ZHN-7-07, diprenylated Okaramine S (**39**), monoprenylated Okaramine T (**40**) and Okaramine U (**41**) deprived of prenyl groups (Table 3, entry 6) [124]. In the cytotoxic evaluation, only diprenylated Okaramine S (**39**) exhibited a promising effect on HL-60 and K562 cell lines with IC_50_ values of 0.78 and 22.4 μM, respectively. Roquefortine F (**42**) and Roquefortine G (**43**) were isolated from a deep ocean sediment derived fungus *Penicillium* sp. and tested on four cell lines: A-549, HL-60, BEL-7402, MOLT-4 (Table 3, entries 7,8) [125]. Roquefortine F (**42**), the more cytotoxic derivative, showed moderate effect with IC_50_ values of 14.0, 33.6, 13.0, and 21.2 μM, respectively, while Roquefortine G (**43**) with IC_50_ values of 14.0, 33.6, 13.0, 21.2 μM, respectively, 42.5, 36.6, >50, and >50 μM, respectively. Fructigenine A (**44**), Fructigenine B (**45**), Rugulosuvine A (**46**) as well as *N*-glycosylated Penicimutanin A (**47**) and Penicimutanin C (**48**) were isolated from a neomycin-resistant mutant 3-f-31 of *Penicillium purpurogenum* G59 [126]. Compounds **44–48** were tested for cytotoxicty using four cancer cell lines: HeLa, BGC-823, MCF-7, K562 and HL-60. While Fructigenine A (**44**), Fructigenine B (**45**), Rugulosuvine A (**46**) exhibited rather weak cytotoxic activity on tested cell lines (IC_50_ > 100 μM) (Table 3, entries 9–11), Penicimutanin A (**47**) and Penicimutanin C (**48**), bearing aglycone attached to the indole ring, showed marked inhibition effect with values IC_50_ [μM] of 10.7 (K562), 6.1 (HL-60), 7.0 (HeLa), 8.3 (BGC-823), and 7.3 (MCF-7) for Penicimutanin A (**47**) and 11.9 (K562), 5.0 (HL-60), 8.6 (HeLa), 8.7 (BGC-823), and 6.0 (MCF-7) for Penicimutanin C (**48**) (Table 3, entries 17,18). Two diastereomeric Eurotiumin A (**49**) and Eurotiumin B (**50**) were isolated from marine-derived fungus *Eurotium* sp. SCSIO F452 and tested for cytotoxicity [127], but did not exhibit a promising effect on SF-268 and HepG2 cell lines up to 100 μM (Table 3, entries 14,15).

#### 4.1.4. Penta- and Hexacyclic Proline-Based DKP

The pentacyclic proline-based DKP are mainly based on series of compounds called the Spirotryprostatins and Cyclotryprostatins, analogs of bicyclic Tryprostatins **21**–**23** (Table 2, entries 1,2). Cui reported the isolation of novel mammalian cell cycle inhibitors, Spirotryprostatin A (**51**) and Spirotryprostatin B (**52**), produced by *Aspergillus fumigatus* strain [128,129]. These compounds, possessing a spiro indole ring system attached to the tricyclic DKP scaffold, inhibited the mammalian cell cycle at G2/M phase with IC_50_ [μM] values of 197.5 and 14.0, respectively (Table 4, entries 1,2) [128,129]. Spirotryprostatin B (**52**) also showed cytotoxic activity on K562 (chronic myelogenous leukemia) and HL-60 cell lines with the MIC values of 35 μg/mL and 10 μg/mL [129] and IC_50_ [μM] value of 14.0 for 3Y1 cell line (rat fibroblasts) [130]. Recently, Spirotryprostatin L (**53**) and its analog **54** [131] were isolated from marine-derived fungus *Penicillium brasilianum*. The authors observed the selective cytotoxicity of **53** and **54** against HL-60 cell line with the IC_50_ values of 6.0 and 7.9 μM, respectively (Table 4, entries 3,4).

Cyclotryprostatin B (**55**), Cyclotryptostatin F (**56**), and Cyclotryptostatin G (**57**) were isolated from *Penicillium brasilianum* together with Spirotryprostatins **53**,**54** (Table 4, entries 1–3) [131]. These compunds exhibited relatively selective cytotoxic effect when tested on breast cancer cell line MCF-7, with IC_50_ values of 5.1, 7.6, and 10.8 μM, respectively. In contrary to Cyclotryprostatins B, F and G (**55**–**57**), Cyclotryprostatin E (**58**), isolated from *Aspergillus sydowii* SCSIO 00305 [132] did not show any cytotoxic effect when tested on A549, A375 (human melanoma) and HeLa cell lines (Table 4, entry 8), which may indicate that the presence of the prenyl group in the structure of the compound is necessary for the appearance of cytotoxic activity. Pentacyclic Versicamide G (**59**), bearing 11-membered lactam ring, was isolated from marine-derived fungus *Aspergillus versicolor*, but was not active against HeLa, HCT-116, HL-60 and K562 cell lines (Table 4, entry 9) [133]. Together with Versicamide G (**59**), six new hexacyclic Versicamides A–F (**60**–**65**) (Table 4, entries 10–15) were isolated and tested for cytotoxicity against HeLa, HCT-116, HL-60 and K562 but significant effect was not observed. Versicamide G (**59**) was further treated with methyl iodide, in the presence of sodium carbonate in tetrahydrofurane, which led to the formation of Versicamide H (**66**) (Table 4, entry 16). In contrary to Versicamides A-G (**59–65**), compound **66** surprisingly exhibited moderate cytotoxic effect with IC_50_ [μM] values of 19.4 (HeLa), 17.7 (HCT-116), 8.7 (HL-60) and 22.4 (K562). When tested on the panel of 18 selected protein kinases, Versicamide H (**66**) showed effective activity on c-Kit (a transmembrane protein that functions as a tyrosine kinase receptor) yielding an inhibitory rate of 60% at a final concentration of 10 μM [133]. In 2019, Li reported isolation an identification of novel hexacylic asperversiamides I−P from a soil-derived fungus *Aspergillus versicolor* [134], and observed that one compound, Asperversiamide I (**67**), exhibited marked cytotoxic activity against HeLa cell line with IC_50_ = 7.3 μM (Table 4, entry 17). Finally, Speramide B (**68**) (Table 4, entry 18), a new prenylated indole alkaloid isolated from the freshwater-derived fungus *Aspergillus ochraceus* KM007 [135] was tested for cytotoxicity on PC3, DU145 (human prostate carcinoma) and LNCaP cell lines, but did not exhibit any effect up to IC_50_ = 40 μM.

#### 4.1.5. Hepta-, Polycyclic and Dimeric Proline-Based DKP

Heptacyclic Speramide A (**69**), isolated together with Speramide B (**68**), from fungus *Aspergillus ochraceus* KM007 [135] was also tested for cytotoxicity on PC3, DU145 and LNCaP, but also did not exhibit any effect up to IC_50_ = 40 μM (Table 5, entry 1). Heptacyclic Stephacidin A (**70**) was isolated from *Aspergillus ochraceus* WC76466 strain [136] and tested on ten cancer and one reference non-cancerous cell lines: PC3 (prostate, testosterone-independent), LNCaP (prostate testosterone-sensitive), A2780 (ovarian parental), A2780/DDP (ovarian mutp53/bcl2+), A2780/Tax (ovarian taxol-resistant), HCT116 (colon parental), HCT116/mdr+ (overexpress mdr+), HCT116/topo (colon resistant to etoposide), MCF-7 (breast estradiol-sensitive), SKBR3 (breast estradiol-independent), and reference LX-1 (non-cancerous, human hepatic stellate cell line). Compound **70** exhibited a strong cytotoxic effect with IC_50_ [μM] values varied from 1.00 (LNCaP) to 13.10 (HCT116/topo) (Table 5, entry 2). Drimentine I (**71**), isolated from *Streptomyces* sp. CHQ-64 strain, was evaluated in vitro for its cytotoxicity against two human tumor cell lines (A549 and HeLa) and exhibited weak activity against human cervical carcinoma cell line HeLa, with IC_50_ values of 16.73 μM (Table 5, entry 3) [137]. Gartryprostatin A (**72**) and Gartryprostatin B (**73**) were isolated and identified as secondary metabolites of *Aspergillus* sp. GZWMJZ-258, an endophyte of the medicinal and edible plant *Garcinia multiflora* [138]. Compounds **72** and **73** were tested for cytotoxic effect on four cancer cell lines: MV4-11, K562, HL-60, and A549 and exhibited selective cytotoxic effect against leukemic MV4-11 cell line with IC_50_ values of 7.2 μM and 10.0 μM, respectively (Table 5, entries 4,5). (+)-Avrainvillamide (**74**), isolated from the fermentation broth *of Aspergillus ochraceus* [87], turned out to be a potent inhibitor of tumor-associated protein—nucleoplasmin [139,140] overexpressed in many human tumors, exhibiting strong cytotoxic effect against HeLa (IC_90_ [μg/mL] = 1.1) [87], T-47D (breast cancer, GI_50_ [μM] = 0.33), and LNCaP (GI_50_ [μM] = 0.42) cell lines [139] (Table 5, entry 6). Its enatiomer **75** exhibited a weaker, but still potent cytotoxic effect, when tested on T-47D and LNCaP cell lines with GI_50_ [μM] values of = 0.91 and 1.4, respectively (Table 5, entry 7) [140]. (+)-Avrainvillamide (**74**) was further investigated as a potential antileukemic agent [140] and tested on five acute myeloid leukemia (AML) cell lines: NB4, HL-60, MV4-11, OCI-AML3, and Molm-13 giving IC_50_ [μM] values from 1.1 (NB4) to 0.078 (Molm-13). Waikikiamide A (**76**) and Waikikiamide B (**77**) were isolated from a Hawaiian marine fungal strain *Aspergillus* sp. FM242. Their cytotoxic effect was evaluated on four cancer cell lines HT1080, (fibrosarcoma), PC3, Jurkat (acute T cell leukemia), and A2780S (human ovarian cancer) [141].

The more potent Waikikiamide A (**76**) exhibited activity with IC_50_ [μM] values from 0.519 (HT1080) to 1.855 (PC3) (Table 5, entry 8), while Waikikiamide B (**77**) from 1.127 (A2780S) to 1.805 (PC3) (Table 5, entry 9). Dimeric Stephacidin B (**78**) was isolated together with Stephacidin A (**70**) from *Aspergillus ochraceus* WC76466 strain [136] and tested on a similar panel of ten cancer and one reference non-cancerous cell lines: PC3, LNCaP, A2780, A2780/DDP, A2780/Tax, HCT116, HCT116/mdr+, HCT116/topo, MCF-7, SKBR3 and LX-1. Compound **78** exhibited stronger cytotoxic effect then compound **70** with IC_50_ [μM] values varied from 0.06 (LNCaP) to 0.46 (HCT116, HCT116/mdr+) (Table 5, entry 10). Nandaseseazine A (**79**) Naseseazine B (**80**) were obtained from *Streptomyces* sp. (CMB-MQ030) isolated from a Fijian marine sediment and tested for cytotoxicity on four cancer cell lines: AGS (gastric adenocarcinoma), SH-SY5Y (neuroblastoma), TF-1 (erythroleukemia) and HT-29 (colorectal adenocarcinoma) but were found to be rather non-toxic compounds [142] (Table 5, entries 11,12). Asperflocin (**81**), an asymmetric diketopiperazine dimer from the sponge-associated fungus *Aspergillus versicolor* 16F-11, and its diastereomer WIN 64821 (**82**) were evaluated as potential anticancer agents on the panel of four cancer cell lines; HT-29, A375, MCF-7, and HepG2, but only Asperflocin (**81**) exhibited moderate selectivity against A375 cell line with IC_50_ [μM] value of 10.29 ± 2.37 (Table 5, entries 13,14) [143].

#### 4.1.6. Sulfur-Containing Proline-Based DKPs

There are numerous examples of various proline-based diketopiperazine alkaloids which have in their structure from one to four sulfur atoms in the form of sulfide or polysulfide bridges or thiomethoxy groups. The influence of the presence of sulfur atoms as well as the type of sulfur-containing functional groups on biological properties of proline-based DKPs is best seen on the example of the tricyclic gliotoxin (**83**) and its analogs [144]. Gliotoxin (**83**), possessing tricyclic structure with disulfide bond, as well as its acetylated derivative—Acetylgliotoxin (**84**) are strong cytotoxic agents when tested on cancer cell lines: SF-268, MCF-7, NCI-H460, and HepG-2 with IC_50_ [μM] varied from 0.08 (MCF-7) to 0.25 (SF-268, NCI-H460) for **83** and from 0.22 (MCF-7) to 0.49 (HepG-2) (Table 6, entries 1,2). 6-deoxy-5*a*,6-didehydrogliotoxin (**85**) also possessing disulfide bond within its structure, also exhibited significantly but slightly weaker cytotoxicity with IC_50_ [μM] varied from 0.68 (MCF-7) to 1.52 (HepG-2) [144] (Table 6, entry 3). Bisdethiobis(methylthio)gliotoxin (**86**), 6-acetylbisdethiobis(methylthio)gliotoxin (**87**) and Dichotocejpin A (**88**), bearing two (**86**,**87**) or one (**88**) thiomethoxy groups instead of disulfide bond, exhibited markedly weaker (**87**,**88**) or no (**86**) cytotoxic effect on tested cell lines [144] (Table 6, entries 4–6). Structural analog of Gliotoxin (**83**) devoid of sulfur groups, Dichotocejpin B (**89**) lacking any activity on above cell lines tested (Table 6, entry 7). From these results it can be generally assumed that the compounds with the disulfide bridge show the greatest cytotoxic effect, the compounds with thiomethoxyl groups instead of the disulfide bridge show a weaker effect, and the compounds devoid of sulfur groups have the weakest cytotoxic effect in the whole series of compounds. Sun [145] reported the isolation and identification of gliotoxin and its analogs from a marine-derived fungus, *Penicillium* sp. including **83**, **85**, **86**, bis(dethio)-10*a*-methylthio-3*a*-deoxy-3,3*a*-didehydrogliotoxin (**90**), bis(dethio)bis-(methylthio)-5*a*,6-didehydrogliotoxin (**91**), 5*a*,6-didehydrogliotoxin (**92**) and Gliotoxin G (**93**) and tested them for cytotoxic effect on P388 cell line (Table 6, entries 1,3,4,8–11). He found that Gliotoxin G (**93**), possessing tetrasulfide bond, was the most active of the tested compounds, exceeding its cytotoxicity of the reference compound gliotoxin (**83**) (IC_50_ [μM] = 0.020 vs. 0.024, Table 6, entries 1,11). On the other hand, bis(dethio)-10*a*-methylthio-3*a*-deoxy-3,3*a*-didehydrogliotoxin (**90**) showed the weakest cytotoxic effect on the line tested with IC_50_ [μM] value of 3.4 (Table 6, entry 8). A series of gliotoxin derivatives including **83**, **84**, **86**, **87** together with reduced gliotoxin (**94**) and 6-deoxy-bis(dethio)bis-(methylthio)-5*a*,6-didehydrogliotoxin (**95**) were isolated from the marine fungus *Neosartorya pseudofischeri* [146] and tested for cytotoxic effect on three cell lines: HEK293 (human embryonic kidney), HCT-116, and RKO (a poorly differentiated colon carcinoma cell line). Compounds **87** and **95** proved to be inactive (Table 6, entries 5,13), compound **86** exhibited moderate inhibition effect with IC_50_ [μM] of 16.39 (HEK293), 8.59 (HCT-116), and 10.32 (RKO) (Table 6, entry 4). Compounds **83**, **84** and **94** showed the strongest cytotoxic effect (Table 6, entries 1,2,12), and reduced gliotoxin (**94**) showed slightly more potent cytotoxicity and selectivity than the reference compound gliotoxin (**83**) (IC_50_ [μM] = 1.58 vs. 1.26 for HEK293, 1.24 vs. 0.46 for HCT-116 and 0.8 vs. 0.41 for RKO). Compounds **83** and **91** were isolated by Zhao from *Aspergillus fumigatus* strain [147], who reported the IC_50_ [μg/mL] values for tsFT210 cell line of 0.15 and 89, respectively (Table 6, entries 1,9). Wang observed the cytotoxic effect of **83**, **86** and **91** on U-937 (histiocytic lymphoma) and PC-3 cell lines [148] and reported IC_50_ [μM] values for U-937 cell line of 0.20 (**83**), 0.52 (**86**) and >100 (**91**) (Table 6, entries 1,4,9). The PC-3 cell line turned out to be more resistant to the tested compounds, for which the IC values were 0.39 (**83**), 15.8 (**86**) and >100 (**91**), respectively [148] (Table 6, entries 1,4,9). Zhao reported [149] that 3-dehydroxymethylbisdethio-3,10*a*-bis(methylthio)gliotoxin (**96**) and two other gliotoxin analogs **86** and **91** could inhibit B16 cancer cell line with inhibitory rate of 86%, 82%, 78%, respectively, at the concentration of 500 μg/mL (Table 6, entries 4,9,14). Coculture of a mine drainage-derived *Sphingomonas* bacterial strain, KMK-001, and a mine drainage-derived *Aspergillus fumigatus* fungal strain, KMC-901, resulted in isolation of two novel analogs of gliotoxin (**83**), glionitrin A (**97**) [150] glionitrin B (**98**) [151], possessing aromatic nitro group within their structures (Table 6, entries 15,16). Glionitrin A (**97**) was tested for cytotoxic effect against six human cancer cell lines, it showed submicromolar inhibition of cell proliferation in the HCT-116, A549, AGS (gastric adenocarcinoma), and DU145 cell lines (IC_50_ = 0.82, 0.55, 0.45, and 0.24 μM, respectively). Glionitrin A (**97**) showed relatively weaker inhibition of the cell proliferation of the MCF-7 and HepG2 cell lines (IC_50_ = 2.0 and 2.3 μM, respectively) [150]. In contrast to glionitrin A (**97**), glionitrin B (**98**) was not cytotoxic against the human prostate cancer cell line DU145 [151]. However, compound **98** caused suppression of DU145 cell invasion, producing 46% inhibition at 60 μM [151]. Phomazine B (**99**), a novel gliotoxin (**83**) analog was isolated from the marine-derived fungus *Phoma* sp. OUCMDZ-1847 [152]. Compound **99** showed a rather moderate/weak cytotoxic effect when tested on HL-60, HCT-116, A549, K562, MGC-803 (human gastric cancer) cell lines with the highest cytotoxicity on MGC-803 (IC_50_ = 8.5 μM) (Table 6, entries 17). As gliotoxin (**83**) has promising cytotoxic properties and a broad spectrum of activity against various types and types of cancer, its structure was also an inspiration for the design and synthesis of a number of bicyclic analogs with a disulfide bridge that showed significant cytotoxic effects [153].

A series of five, tetracyclic, proline-based DKPs, containing disulphide and trisulphide bridges, were isolated by Wang from deep sea-derived fungus *Acrostalagmus luteoalbus* SCSIO F457 [154]. Luteoalbusin A (**100**), Luteoalbusin B (**101**), T988A (**102**), Gliocladine C (**103**) and Gliocladine D (**104**) were tested for cytotoxicity on four cancer cell lines: SF-268, MCF-7, NCI-H460, and HepG-2, exhibiting significant effect, where MCF-7 cell line proved to be the most sensitive and NCI-H460 line the most resistant to the compounds tested (Table 6, entries 18–22). The highest cytotoxicity was observed for Luteoalbusin A (**100**) with IC_50_ [μM] values of 0.46 (SF-268), 0.23 (MCF-7), 1.15 (NCI-H460), and 0.91 (HepG-2), while the weakest cytotoxicity was for Gliocladine D (**104**) with IC_50_ [μM] values of 2.49 (SF-268), 0.65 (MCF-7), 17.78 (NCI-H460), and 2.03 (HepG-2). Tetracyclic, chlorinated Sporidesmin A (**105**) was isolated from a Freshwater *Delitschia* sp. [155] and was evaluated against the African American prostate cancer cell line (E006AA-hT) under hypoxic conditions giving IC_50_ [μM] value of 2.5 (Table 6, entry 23). A series of six, pentacyclic, proline-based DKPs, containing disulphide bridge were isolated by Meng from *Penicillium brocae* MA-231, an endophytic fungus derived from the marine mangrove plant *Avicennia marina* [156]. Brocazines A-F (**106**–**111**) were evaluated for possible cytotoxic effect using nine cancer cell lines: Du145, HeLa, HepG2, MCF-7, NCI-H460, SGC-7901 (gastric cancer), SW1990 (pancreatic adenocarcinoma), SW480 (colon adenocarcinoma), and U251. The authors reported that two compounds, Brocazine C (**108**) and Brocazine D (**109**), did not exhibit significant cytotoxic activity against the tested cancer cell lines with IC_50_ values greater than 20 μM (Table 6, entries 26, 27). Brocazine A (**106**) exhibited the cytotoxic effect in the range of 2.0 (SW480)—6.8 (HeLa) (Table 6, entry 24), Brocazine B (**107**) in the range of 1.2 (SW480)—6.4 (SW1990) (Table 6, entry 25), Brocazine E (**110**) in the range of 2.1 (SW1990)—12.4 (NCI-H460) (Table 6, entry 28) and Brocazine F (**111**) in the range of 0.89 (NCI-H460)—8.0 (SGC-7901) (Table 6, entry 29). Pentacyclic, proline-based DKPs, possessing disulphide bridge or thiomethoxy groups were isolated by Kong from the marine-derived fungus *Phoma* sp. OUCMDZ-1847 [152]. Epicorazine A (**112**), Epicorazine B (**113**), Epicorazine C (**114**), and Exserohilone A (**115**) were tested for cytotoxic effect against five cancer cell lines: HL-60, HCT-116, A549, K562, and MGC-803 (Table 6, entries 31–34). The authors found that Epicorazine A (**112**) turned out to be the most cytotoxic from the compounds tested with IC_50_ values of 0.05 (HL-60), 0.33 (HCT-116), 2.3 (A549), 1.5 (K562), and 2.7 μM (MGC-803) (Table 6, entry 31), and HL-60 cell line was the most susceptible from the lines used in the research. Brocazine G (**116**), Spirobrocazine A (**117**), and Spirobrocazine B (**118**) were isolated from the mangrove-derived endophytic fungus *Penicillium brocae* MA-231 and tested for cytotoxicity against sensitive and cisplatin-resistant human ovarian cancer cell lines A2780 and A2780 CisR (Table 6, entries 35–37) [157], whereas Spirobrocazine A (**117**) and Spirobrocazine B (**118**) did not exhibit any activity, Brocazine G (**116**) showed strong activity not only to A2780 but also to A2780 CisR cells, with IC_50_ values of 0.664 and 0.661 μM, respectively, stronger than that of cisplatin used as a positive control. 5′-Hydroxy-6′-ene-epicoccin G (**119**), 7-methoxy-7′-hydroxyepicoccin G (**120**), 8′-acetoxyepicoccin D (**121**), 7′-demethoxyrostratin C (**122**), Epicoccin E (**123**), Epicoccin G (**124**), and Rostracin C (124) were isolated by Chi from the deep sea-derived fungus *Epicoccum nigrum* SD-388 and tested for cytotoxicity against Huh7.5 liver tumor cells (Table 6, entries 38–44) [158]. Compounds **119–121**, **123**, and **124** turn out to be completely inactive, while 7′-demethoxyrostratin C (**122**) and Rostracin C (**125**) showed marked activity with IC_50_ values of 9.52 and 4.00 μM, respectively, but were also toxic to human normal liver LO2 cell line. The authors suggested that disulfide bridge is likely required for cytotoxic activity. Chinworrungsee reported isolation and identification of three novel pentacyclic compounds **126–128** obtained from seed fungus *Menisporopsis theobromae* BCC 3975, which were tested against three cancer cell lines: KB (papilloma), BC-1 and NCI-H187 (lung carcinoma) (Table 6, entries 45–47) [159]. Compounds **126–128** exhibited rather weak cytotoxic effect with the highest IC_50_ values for **126** of 22.9 (NCI-H187) and 29.2 μM (BC-1) and for **127** of 20.3 μM (NCI-H187). A number of dimeric, sulfur-containing DKPs were isolated from fungal sources including Cristazine (**129**), found in the mudflat-sediment-derived fungus *Chaetomium cristatum* [160]. Cristazine (**129**), exhibited cytotoxic effect on HeLa and A451 with IC_50_ values of 0.5 μM, respectively, induced apoptosis via the death receptor pathway [160,161] (Table 6, entry 48). Chaetocin (**130**) is a dimeric sulfur-containing DKPs, which exhibited a wide range of cytotoxic effect on both solid and blood tumors [169,170]. Lai observed that Chaetocin (**130**) induced differentiation and shows synergistic cytotoxicity with other epigenetic drugs in acute myeloid leukemia cells HL-60, U937 and KG-1a [162]. He reported IC_50_ values of 0.153 (HL-60), 0.096 (U937) and 0.103 (KG-1a) (Table 6, entry 49). A series of sulfur-containing, dimeric DKP were isolated from Bionectriaceae cultures MSX 64546 and MSX 59553, including Verticillin A (**131**), Verticillin H (**132**), Sch 52900 (**133**), Sch 52901 (**134**), Gliocladicillin A (**135**), Gliocladicillin C (**136**), and 11′-Deoxyverticillin (**137**) [163]. All compounds (**131**–**138**) were evaluated for cytotoxicity against a panel of human cancer cell lines (HT-29, H460, SF-268, MCF-7 and MDAMB-435), displaying IC_50_ values ranging from 1.2 mM to 10 nM (Table 6, entries 50–57). Verticillin A (**131**), Verticillin H (**132**), and their ester derivatives also exhibited strong cytotoxic effect on MDA-MB-231 and OVCAR3 cell lines (Table 6, entries 50–51), in the few cases, some of the ester derivatives showed higher cytotoxicity than the parent compounds [164]. Further studies proved the effectiveness of Verticillin A (**131**) in human gastric (AGS) [165], cervical (HeLa) [165], and ovarian cancer [166] cells lines (Table 6, entry 50). 11′-Deoxyverticillin (**137**) and 11,11′-dideoxyverticillin (**138**) were also very active against HCT-116 human colon carcinoma (IC_50_ = 0.030 μM/mL) (Table 6, entries 56–57) [167] and 11,11′-dideoxyverticillin (**138**) also showed strong cytotoxic effect against human breast adenocarcinoma MDAMB-468, MCF-7, MDA-MB-435, and MDA-MB-231 cell lines with IC_50_ values of 0.281, 0.158, 0.223, and 0.138 μM, respectively [168] (Table 6, entry 57).

Except of the direct cytotoxic effect, there are some other reports suggesting usefulness of proline-based DKPs in the treatment of cancer and supporting anticancer therapy. Yu reported [171] that several pentacyclic, hexacyclic, or dimeric proline-based DKPs could be potent inhbitors of BRD4 protein (Bromodomain-containing protein 4), very often expressed in various types of tumors. Fumitremorgin C (**139**) and their analogs are selective and potent inhibitors of the multidrug-resistance protein (BCRP/ABCG2) that mediated resistance to chemotherapeutics [172]. Octacyclic proline-based DKP, Nocardioazine A (**140**) proved to be a noncytotoxic inhibitor of the membrane protein efflux pump P-glycoprotein, reversing doxorubicin resistance in multidrug resistant colon cancer cells [173]. Gliotoxin analogs could also act as non-histone arginine methyltransferase inhibitors [174]. Finally, Leptosins, proline 2,5-DKP derivatives, isolated from marine fungus *Leptoshaeria*, play a role of antitumor agents inhibiting DNA topoisomerases [175].

### 4.2. Other Bio-Activities

In the following sub-sections, key issues are only discussed briefly since a growing potential in terms of broad spectrum bio-activities of proline-based DKPs have been observed only recently.

#### 4.2.1. Neuroprotection

Neurodegenerative diseases, such as Alzheimer’s disease (AD), Parkinson’s disease (PD), and amyotrophic lateral sclerosis (ALS) are age-dependent multifactorial pathologies [19] with various etiopathogenetic sources, but sharing common pathogenic mechanisms: neuroinflammation, oxidative and ER stress; all characterized by neuronal death and degeneration leading to a progressive functional decline. There is no currently available treatment to cure any of those diseases with only symptomatic or slowing down the progress of mental regression is offered. 

Therefore, there is a demand for the new class of pharmaceutics providing the neuroprotection and preventing neurons damage. Neurodegenerative diseases could be treated DKP, and their analogs, as they can cross the BBB, because of their unique cyclic structure, and inhibit neuronal degeneration due to their good pharmacokinetic profile and long-lasting neuroprotection. There are mainly two classes of DKP: 1., the TRH-related and 2., unsaturated compounds, proposed recently as potential drug candidates possessing remarkable neuroprotective profile.

##### TRH-Related DKP

Naturally occurring hormone, thyrotropin-releasing hormone (TRH), l-pyroglutamyl-l-histidyl-l-prolineamide, was the first characterised as playing neuromodulatory role within the central nervous system (CNS) with noticing its rapid enzymatic degradation, what is a big disadvantage for clinical use [176,177]. Therefore, during the last two decades researchers have focused on the cyclic derivatization of the linear TRH peptides, naturally occurring and synthetic [19] with the most investigated cyclo(His-Pro), (CHP), produced by the metabolism of TRH as regulatory agent mediating brain activities [176,177]. CHP is ubiquitous in the cerebrospinal fluid (CSF), blood, and several other body fluids [178] playing important roles in pain awareness, body core temperature, food intake, and modulating prolactin secretion, acting as an endocrine effector. There is a correlation between the decrease CHP level and increased age of patients reaching a value in adulthood of 300 pg/mL in adult age, while for the preterm babies it is 1909 pg/mL. CHP and CHP-like compounds, when increased the presence in CSF, they can improve cognitive function and enhance neurological recovery after trauma due to their molecular mechanisms of neuroprotection to be related to apoptosis and necrosis; DNA repair; oxidative stress; ER stress and unfolded protein response; mitochondrial energy metabolism; together with immunity pathways. Additionally, CHP was found to protect neurons from cytotoxicity induced by salsolinol, a metabolite of l-DOPA linked to Parkinson. CHP was proven to inhibit the pro-inflammator NF-kappaB pathway and its isomers were tested against neuropathological conditions showing an ameliorating potential via elevating METAP2 (responsible for promoting cell proliferation in SH-SY5Y neuroblastoma cells and for exhibiting immune-modulatory activity) expression decreased by Abeta 1–42 [179]. Moreover, the CHP isomer exhibited noncytotoxic and non-mutagenic features in cultured human blood cells and DKP derivatives are promising therapeutics against neurodegeneration-related pathologies [19,179]. They are multipotent anti-Alzheimer drug candidates [179,180]. They also are promising in the treatment of other neurodegenerative diseases, such as amyotrophic lateral sclerosis (ALS, mentioned previously) [179]. The TRH-related DKPs are reported to prevent or reduce both necrotic and apoptotic cell death, they were tested in diverse in vitro models to present significantly improved cognitive and motor outcomes and to reduce lesion volumes following traumatic brain injury. From all the bio-evaluations reported in the literature, it could be concluded that the proline residue is a structural element contributing significantly to the DKPs biopotency [19].

##### Unsaturated DKPs

DKPs consisting of unsaturated units possess the ability to enhance somatosensory-motor function and long-term histological outcome to exhibit a significant radical scavenging activity. The naturally occurring and synthetic DHP derivatives are comprehensively described by Pinnen and colleagues in their mini-review [19]. They concluded that in those DKP scaffolds, the unsaturated motifs play a pivotal role in the protective and reparative properties of these DKPs, suggesting that the increase of bioactivity is strictly related to the presence of unsaturated groups in the DKP and such design can contribute significantly to the development of the potential therapeutic intervention in neurodegenerative diseases.

##### Other DKPs

The cyclic dipeptides not structurally related to TRH or with unsaturated fragment have also been shown to exhibit neuroprotective and nootropic properties. The first example is cyclo(Leu-Gly), the cyclic analog of H-Pro-Leu-Gly-NH2, the *C*-terminal neurohypophyseal tripeptide of oxytocin, reducing the blockade of memory induced by puromycin, a protein synthesis inhibiting drug which can impair memory processes. The second example is cyclo(Pro-Gly) that was found to reduce motor neuron death induced by glutamate, probably due to the presence of neuroprotective Gly in the structure and potentially slowing down the progression of ALS [181].

Cyclo(Pro-Gly) is the only, apart from cyclo(His-Pro), cyclic dipeptide in the central nervous system. It can be derived from glycine-proline-glutamate. It has anxiolytic activity [181]. It is an endogenous prototype of Piracetam (derivative of gamma-aminobutyric acid), a nootropic drug. Cyclo(Pro-Gly) and its analog cyclo-l-glycyl-l-2-allylproline are neuroprotective after ischaemic brain injury [182]. Moreover, it has the antinociceptive effect that is mediated through the interaction with opioid system, with fewer side effects [183]. Cyclic glycyl proline compounds are promising in the treatment of Pitt Hopkins Syndrome, a rare, genetic neurological disorder [184], for which there is no cure so far. Moreover, cyclo(Pro-Gly) is a nature nutrient of the blackcurrant anthocyanins. It can improve the function of insulin-like growth factor-1, in Parkinson disease [185,186]. In the latter case, cyclo-l-glycyl-l-2-allylproline has relevance as well [187]. Interestingly, cyclo(Gly-Pro) is a major bioactive component of *raksi*, an ethnic beverage, considered as a remedy for every possible disorder related to high-altitude sicknesses. This cyclo-dipeptide, reported earlier as antibiotic, is also a potential antioxidant, reported as promising anti-inflammatory, pain reliever, neuroprotective, and antinociceptive [27,188,189]. Cyclo(l-His-l-Pro) and cyclo(l-Trp-l-Pro) are anti-depressants and learning motivation improvers [190]. 

The derivatives of gliotoxin, without a disulfide bridge, are a new promising class of opioid antagonists with neuroprotective properties. Other DKP-based compounds are inhibitors of opioid receptors as well. DKPs show clinical implication in neurological recovery after a traumatic brain or spinal cord injuries [191,192,193]. 

2,6-DKPs display a broad anticonvulsant activity in epilepsy, a major neurological disorder, exhibiting a broad spectrum of seizure-suppressing effect. Notably, traditional antiseizure drugs lead to severe side effects and drug–drug interactions caused by enzyme induction. Keeping in mind the drug-resistance as growing problem, DKPs are a good promise for the need of either more effective, safer, or ‘disease modifying drugs’ that inhibit or reverse the progression of the sickness [96,194].

##### Mode of Actions

Due to their heterocyclic structure, DKPs possess high biostability against the proteolysis and bioavailability for the active intestinal absorption and crossing the blood-brain barrier (BBB), both challenging targets for natural linear peptides. DKPs can reduce oxidative and ER stress as well as inflammation, the main culprits of several neurological disorders. Some of DKPs can act in the bacterial quorum sensing (QS), therefore they can influence the gut microbiome regulation, quite important in any clinical dysregulations, including neuropathology. This section is divided into four components, each presenting the individual aspect of the DKP properties.

##### The Blood-Brain Barrier Transporters

The drugs delivery to the specific target sites is a challenge. Biological barriers such as blood brain barrier (BBB) prevent the passage of nearly all drugs to the brain [195]. Most of the DKPs have an ability to cross the BBB [74,196]. More specifically, the ring structure allows permeability, as well as protects against proteolytic degradation. Thus, DKPs are perfect candidates for new therapeutics to treat brain diseases by oral administration [19,197]. The DKP-derived template has been also investigated as a brain shuttle for the delivery of medicinal agents with limited ability to cross the BBB, bypassing the limited bioavailability of several drugs [19,196].

##### Antioxidant Activity

DKPs are a source of natural antioxidants [198,199,200] due to their significant radical scavenging ability. The spiro-anthronopyranoid DKPs, namely variecolortides, isolated from the mycelia of the halotolerant fungal strain *Aspergillus variecolor*, exhibit antioxidant activity [4]. The DKPs with the DOPA catechol entity [201], such as cyclo(His-Pro), CHP [176] were isolated from *Bacillus* sp. N strain [202], as an example. CHP reduces oxidative and endoplasmic reticulum stress by increasing antioxidant capacity in the potential treatment of neurodegenerative diseases [179]. The treatment with CHP isomers provided significant positive oxidative alterations via reducing oxidant status and supporting antioxidant capacity in the cellular AD model by protecting rat insulinoma cells (RINm5F) from streptozotocin (STZ)-induced in vitro cytotoxicity by minimizing of nitric oxide (NO) production and lipid peroxidation [179]. The proline structural element in CHP is responsible for its neuroprotective, nootropic, and cognitive-enhancing properties due to its specific conformation [203,204]. The DKP compounds enhance memory function and the incorporation of the proline in the 2,3- or 2,5-DKP skeleton presents a crucial aspect of neuroprotective action of theses cyclic peptides [19]. They are multipotent anti-Alzheimer drug candidates [179,180]. They also are promising in the treatment of other neurodegenerative diseases, such as ALS [179]. 

##### DKP and Gut Microbiota

DKP and their analogs have been isolated from Gram-positive and Gram-negative marine bacteria and identified as a promising source of bioactive compounds with potential pharmacologic activity. These cyclic dipeptides are involved in the bacterial quorum sensing (QS), a cell-to-cell communication system based on the production, secretion, and detection of signals, QS effectors, that contain a family-defining cyclic dipeptide core/scaffold and are produced by proteobacterial species as well as by humans [205]. Therefore, they could play a crucial role in gut dysbiosis, a primary factor behind various gastrointestinal disorders causing increased intestinal and BBB permeability via microbiota-gut-brain axis. Such uncontrolled accumulation of misfolded proteins can facilitate the axonal damage and neuronal demyelination in the pathogenesis of neurodegenerative disorders such as Parkinson’s disease, Alzheimer’s disease, multiple sclerosis, and amyotrophic lateral sclerosis [206].

##### Protein–Protein Interaction Mechanism

The role of DKP derivatives can be especially important in relation to ‘difficult targets’ such as protein–protein interactions. More specifically, DKP scaffolds can play a lead role as antiprion agents [195]. Prion diseases, allo known as transmissible spongiform encephalopathies, are fatal infectious and neurodegenerative disorders, while effective drugs are lacking. Both prion and Alzheimer’s diseases are protein misfolding disorders. The interplay of prion and Alzheimer’s diseases, which cause dementia in over 40 milion people all over the world, revealed an urgent need for antiprion treatment. The progression of the disease is caused by the conformational change of the prion protein, from alpha-helical to beta-sheeted form, forming amyloid aggregates. A planar conformation of DKP core is the main determinant for this kind of bio-activity [207]. DKPs can modulate protein–protein interactions. They can optimize the desired anti-aggregating properties [195].

#### 4.2.2. Metalloproteinase Inhibitors

DKPs, such as cyclo(His-Pro), are important agents in terms of matrix metalloproteinases (and collagenase inhibitors), zinc-based enzymes, engaged in the degradation and remodeling of the extracellular matrix, mainly with regard to cancer or arthritis [32,208], but also multiple sclerosis.

#### 4.2.3. Antibacterial Activity

Novel effective antibacterial agents are extremely needed to fight the increasing serious problem of drug-resistant bacterial infections, claiming the lives of millions of people worldwide [209]. It is the alarming global problem and serious challenge. Short peptides, due to their unique features, are considered as antibiotics of the future [11]. DKPs have great potential in this area due to their specific features (Table 7), mentioned earlier. In particular, proline-based DKP, a known example of DKP classic antibiotic, is bicyclomycin, which is produced by *Streptomyces sapporonensis* and *S. aizunensis* against diarrhea [210,211,212,213]. It is also a rho (a member of the RecA-type ATPase) inhibitor [213]. Cyclo(l-Leu-l-Pro) was found to inhibit the growth of Vancomycin-resistant *Enterococcus faecalis* strains with MIC value of 12.5 μg/mL, an important finding as these types of strains are sometimes untreatable by any current antibiotic or antibiotic combinations [214]. The combination of cyclo(l-Leu-l-Pro) and cyclo(l-Phe-l-Pro) was synergistic with MIC values of 0.25 to 1 μg/mL and was found to be active also against *Micrococcus luteus*, *Escherichia coli, Candida albicans*, *Staphylococcus aureus*, and *Cryptococcus neoformans* with MIC values of 0.25 to 0.5 μg/mL [32,215]. Other antibiotics, such as avrainvillamide, speramide A, fumitremorgin C, tryprostatin B, spirotryprostatin, are active against multidrug-resistant bacteria [32,41]. Cyclo(l-Trp-l-Pro), known as brevianamide F, isolated from marine fungi *Aspergillus fumigatus* and *Aspergillus* sp. and from the bacterium *Streptomyces* sp., has activity against Gram-positive bacteria [32,216]. Brevianamide S, dimer cyclo(Pro-Trp)-cyclo(Pro-Trp), with antimycobacterial effect, should not be overlooked. Verpacamides, e.g., cyclo(l-Arg-l-Pro), have both antimicrobial and anticancer activity [217]. In another report, seven DKPs from the fermentation broth of *Aspergillus fumigatus* isolate, apart from cyclo(l-Leu-l-Pro) and cyclo(l-Phe-l-Pro), cyclo(l-Gly-l-Pro), cyclo(l-Pro-l-Pro), cyclo(l-Pro-l-Val), cyclo(l-Leu-l-trans-4-OH-Pro), and cyclo(l-Phe-l-trans-4-OH-Pro), were found to exhibit weak antibacterial activity inhibiting the growth of *Staphylococcus aureus* and *Micrococcus luteus* [218]. Additionally, cyclo(l-Pro-l-Pro) isolated from *Lucilia sericata* demonstrated antibacterial activities against *Pseudomonas aeruginosa* and *Micrococcus luteus* [219]. Five DKPs possessing the d-configuration for all amino acids, cyclo(d-Phe-d-Pro), cyclo(d-Leu-d-Pro), cyclo(d-Pro-d-Val), cyclo(d-Ile-d-Pro), and cyclo(d-Phe-trans-4-OH-d-Pro), isolated from bacteria in larvae of mollusks, exhibited strong antibiotic activity against *Vibrio anguillarum*. Structure-activity relationship studies revealed that at least one d-amino acid was required for antibacterial activity and that even better MIC was obtained in the case of d-, d-enantiomers [220]. Finally, the in vitro synergistic antibacterial activity of six proline-based DKPs, cyclo(d-Pro-l-Phe), cyclo(l-Pro-l-Met), (cyclo(d-Pro-l-Leu), cyclo(l-Pro-l-Phe), cyclo(l-Pro-l-Tyr), and cyclo(l-Pro-d-Tyr), and cyclo(d-Pro-l-Tyr)) was investigated in combination with two different antibiotics. The results showed synergistic interaction with imipenem, whereas a combination of certain cyclic dipeptides with ceftazidime was additive, suggesting a potential role in delaying the development of resistance. The cytotoxicity of cyclic dipeptides was evaluated against VERO cell line (African Green Monkey kidney cell line), and no cytotoxicity was recorded for concentrations up to 100 μg/mL [202,221].

Among antibacterial agents, DKP alkaloid—actinozine A, cyclo(2-OH-Pro-l-Leu), but also cyclo(2-OH-d-Pro-l-Leu), cyclo(d-Pro-l-Phe), and cyclo(l-Pro-l-Phe) can be mentioned as well [222,223].

In addition, cyclo(l-Pro-l-Leu) isolated from *Aspergillus aculeatus* has antibacterial activity, mainly against *Escherichia coli* [224].

Moreover, indole DKP alkaloids, such as spirodesmin A and verruculogen, should not be overlooked [57].

##### Quorum-Sensing Agonists and Antagonists

In regard to quorum sensing, proline- and hydroxyproline-based DKPs were found to influence cell–cell signaling, offering alternative ways of biofilm control by interfering with microbial communication [225,226,227,228,229]. Cyclo(l-Leu-l-Pro) and cyclo(l-Pro-l-Tyr) found in *Pseudomonas putida*, were able to activate the biosensor strain *Agrobacterium tumefaciens* NT1 [221,230]. Holden et al. reported the activation of quorum sensing in *Chromobacterium violaceum* by cyclo(l-Pro-l-Val) [39]. In another study, cyclo(l-Phe-l-Pro) produced by *Vibrio vulnificus* was found to activate the quorum sensing on an *E. coli* biosensor strain by modulating the expression of the gene encoding OmpU protein [231]. Finally, cyclo(l-Phe-l-Pro) and cyclo(l-Pro-l-Tyr) produced by *Lactobacillus reuteri*, were reported to interfere with quorum sensing in the Gram-positive *S. Aureus* [41,221,227,232,233,234].

#### 4.2.4. Antifungal Activity

Many proline-based DKPs have been characterized for antifungal indications. For instance, cyclo(l-Gly-l-Pro), cyclo(l-Arg-d-Pro), cyclo(l-Arg-l-Pro), cyclo(l-His-l-Pro), and cyclo(l-Pro-l-Tyr) were reported to inhibit the growth of *Saccharomyces cerevisae* [235]. Additionally, Kumar et al. reported the isolation of proline-containing DKPs, namely cyclo(d-Pro-l-Leu), cyclo(l-Pro-l-Met), cyclo(l-Pro-l-Phe), cyclo(d-Pro-l-Phe), cyclo(d-Phe-l-Pro), cyclo(d-Pro-d-Phe) and cyclo(l-Pro-d-Phe) isomers, as well as cyclo(l-Pro-l-Tyr), and cyclo(l-Pro-d-Tyr) from a culture filtrate of a *Bacillus* bacterial strain associated with an entomopathogenic nematode. These cyclic dipeptides showed significantly higher activity than the commercial fungicide bavistin against *Fusarium oxysporum, Rhizoctonia solani*, and *Pencillium expansum* [227,236,237]. Cyclo(l-Pro-l-Leu), cyclo(d-Pro-l-Leu), and cyclo(d-Pro-l-Tyr) displayed superior antifungal activity against *Aspergillus flavus*, *Candida albicans*, and *F. oxysporum* [41]. Cyclo(l-Met-l-Pro), isolated from *P. aeruginosa*, and cyclo(l-Leu-l-Pro) is an antifungal agent isolated from *Streptomyces* sp. and the bacterium *Achromobacter xylosoxidans* with activity against diverse fungi [32]. Another example of antifungal against is cyclo(l-Phe-trans-4-OH-l-Pro) [40]. Moreover, DKP derivatives, such as cis-cyclo(Pro-3-chloro-Tyr) and trans-cyclo(Pro-3-chloro-Tyr) from marine sediments in Aegean and Ionian seas can be mentioned too [18].

Moreover, DKPs, especially those containing proline, as chitinase inhibitors, can act as insecticides, fungicides, and antimalarials. DKPs are chitinase inhibitors. Chitin, polymer of *N*-acetylglucosamine, is a key element of diverse pathogens (fungal, nematodal, and insect). Chitinases, hydrolyzing chitin, play an essential function in the life cycle of pathogens and pathogenesis [32,238,239,240]. Cyclo(l-Arg-d-Pro) and cyclo(l-His-l-Pro), produced by the marine bacteria *Pseudomonas*, inhibit the growth of *Saccharomyces cerevisiae* through mimicking an intermediate structure in the pathological cycle of chitin [241]. It is an appealing lead compound drug [74]. In this context, cyclo(l-Phe-l-Pro) and cyclo(l-Phe-trans-4-OH-l-Pro) are antifungal compounds as well [40].

#### 4.2.5. Antiviral Activity

DKPs containing proline motif exhibit diverse antiviral activities. In this context, there are intriguing indole DPK alkaloids, which are widely distributed secondary metabolites of microorganisms in fungi *Aspergillus, Penicillium, Pestalotiopsis, and Chromocleista*, and have diverse bio-activities [57]. Marine-derived proline-based DKP alkaloids, such as haenamindole, produced by *Penicillium* sp. [14,242], raistrickindole from *P. raistrickii* [113], showing activity against the hepatitis C virus, are good examples. Furthermore, rubrumlines from *E. rubrum* [14], the indolyl DKP derivatives, and aspamide DKPs, isolated from the *Aspergillus versicolor*, endophyte from the sea crab, *Chiromantes haematocheir*, revealing anti-influenza A activity [243] are worthwile too. The latter may be helpful in the treatment of Severe Acute Respiratory Syndrome Coronavirus 2 (SARS-CoV-2) [243]. Moreover, eutypellazines, thiodiketopiperazine-type alkaloids from *Eutypella* sp. present an inhibitory effect in relation to the human immunodeficiency virus (HIV) with low cytotoxicity [244]. Epicoccins isolated from the *Epicoccum nigrum* are another example of proline-containing DKP derivatives causing HIV inhibitory effect [245]. Notably, DKPs are good anti-HIV drugs with high selectivity and few side effects. They inhibit HIV infection and replication [8]. DKPs, as chemokine receptor antagonists, act against multidrug-resistant strains of HIV [41]. Other examples of antiviral DKPs are cis-cyclo(l-Leu-l-Pro), cis-cyclo(l-Phe-l-Pro), and cis-cyclo(l-Val-l-Pro) acting mainly as inhibitors to the proliferation of influenza A virus (H3N2), but also against plant and human pathogenic fungi [246]. Cyclo(l-Tyr-l-Pro) isolated from the Red Sea *Spongia officinalis*, reveals a strong effect as a hepatitis C virus protease inhibitor [247]. It should be mentioned that haenamindole has weak inhibitory activity against the hepatitis C virus with the IC50 value of 76.3 µM [14]. Cyclo(l-Phe-trans-4-OH-l-Pro), the same as cyclo(l-Pro-l-Leu), cyclo(4-*trans*-hydroxy-l-Pro-l-Leu), and cyclo(4-*trans*-hydroxy-l-Pro-l-Phe) isolated from *Streptomyces gougerotii* GT and *Microbulbifer variabilis* exhibited a significant reduction effect on the dengue virus type 2 replication [248].

Furthermore, we can highlight that gliotoxin, mycotoxin containing proline-based DKP core, is an either antiviral or anticancer, antimicrobial, antifungal, and immunomodulatory agent.

Clomycin, albonoursin, ambewelamides, and phenylahistin were reported as agents with antibacterial, antiviral, and immunosuppressive properties [249]. DKP derivatives containing proline moiety play a role of inhibitors of viral haemorrhagic septicemia virus (VHSV), a deadly salmonid rhabdovirus of economic importance in the aquaculture industry [250].

##### Nucleoside Natural Products

Nucleosides, as the parts of DNA/RNA, are important elements for cells and their metabolic pathways, e.g., energy donors, second messengers, enzyme cofactors. Nucleoside natural products are secondary metabolites of microbes. They offer wide range activity, not only antibacterial, antifungal, antiviral, but also antitumor. In this context, proline-containing DKPs are promising for drug discovery, especially antibiotic development [251,252].

#### 4.2.6. Anthelmintic Activity

Drimentines have anthelmintic activity against various parasites [253,254,255]. DKPs demonstrate an antimalarial effect [256,257,258,259]. Naseseazine C, isolated from a marine sediment derived *Streptomyces* sp., as well as DKP from *Metarhizium* sp. fungi, have moderate activity against malaria parasites [260,261]. Gancidin W, isolated from *Streptomyces* sp., showed low-toxicity antimalarial effect [262]. Furthermore, the DKP alkaloids, cyclo(l-Pro-l-Leu), cyclo(l-Pro-l-Phe), and tryprostatin B are active against amastigote-like forms of *Leishmania amazonensis* and intracellular amastigotes of *Leishmania infantum* [263]. They inhibit *Leishmania donovani*, a fly-borne protozoan that causes *visceral leishmaniasis*. It is noteworthy that antileishmanial DKP alkaloids from the endophytic fungus *Trichosporum* sp. also exhibit antimalarial, antifungal, and antibacterial activities [264].

Fumitremorgin B and verruculogen are potential growth inhibitors of the parasites *Leishmania donovani* as well as *Trypanosoma brucei brucei* [57].

#### 4.2.7. Antitoxin Activity

Cyclo(l-Leu-l-Pro) and cyclo(l-Ala-l-Pro) are inhibitors of production of aflatoxin, carcinogenic toxin, produced by *Aspergillus* sp. [265,266]. Notably, proline moiety is critical to inhibitory activity. The cis-trans proline isomerism of the *N*-alkylamide bond in the DKPs is involved in the receptor-mediated bio-activity [267,268]. The relationship between this activity and molecular configuration can be crucial in defining the inhibition mechanism of DKPs [265].

#### 4.2.8. Antagonists of G Protein–Coupled Receptors

G protein-coupled receptors (GPCRs) are the largest family in the genome, implicated in different diseases, such as cancer, Alzheimer’s disease, depression, type 2 diabetes mellitus, obesity etc., and represent therapeutic targets.

Interestingly, the chemokine receptor 5 (CCR5) antagonists, on the basis of 2,5-DKP scaffold, which was developed for the anti-HIV drug Aplaviroc, belong to a GPCR class [32,82]. The potential of DKPs in relation to classical GPCRs is described below.

##### Inhibitors of GABAergic Receptors

GABA (gamma-aminobutyric acid) antagonists are drugs inhibiting the action of GABA inhibitory neurotransmitters in the central nervous system. Cyclo(His-Pro) is a cyclic dipeptide endogenous to the brain due to its structural similarity to tyreotropin-releasing hormone [269]. Its aza-analogs are a promising scaffold in the discovery of next-generation GABA inhibitors [270]. Piperazinediones reveal anxiolytic activity through the modulation of GABAergic transmission [271].

##### Antagonists of Serotoninergic 5-HT1A Receptors

Proline-based arylpiperazine 2,5-DKPs bind with high affinity and selectivity to the 5-HT1A receptor, revealing an improved anxiolytic profile [32,272,273].

##### Antagonists of Oxytocin Receptor

Oxytocin, parathyroid hormone, and alpha-melanotropin have relevance in the occurrence of preterm birth. DKP core facilitates the design and optimization of novel agents to either complement or antagonize the action of the parent hormone [17,23,274,275,276,277,278]. DKP-based retosiban, and epelsiban, more potent than early non-DKP-based atosiban, was previously developed as an oral drug for the prevention of premature labor [278,279,280]. However, proline-based DKPs as potent and selective antagonists provide effective approaches [278,280] in the treatment of sexual dysfunction, e.g., premature ejaculation, either the treatment or prevention of benign prostate hyperplasia is also worthwhile [32,281,282].

#### 4.2.9. PDE-5 Inhibitors

DKPs can act as efficient inhibitors of phosphodiesterase-5 (PDE-5) useful in the treatment of sexual/erectile dysfunction [78,283,284] as a revolutionary approach. A well-known example is tadalafil (and its analogs), which is a highly potent and highly selective drug [79,80,285,286,287] with limited adverse effects [288] and a prolonged action [289,290]. Notably, tadalafil has the longest half-life of the PDE5 inhibitors.

#### 4.2.10. PAF Inhibitors: Anti-Inflammatory Activity and Beyond

PAF, a d-glycerol-derived phospholipid, a potent endogenous mediator of inflammation, generated via diverse cells, evokes its bio-activity through interaction with specific G-protein coupled receptors, platelets, neutrophils, and other inflammatory cells [32,291,292,293]. PAF has relevance in diverse dysfunctions, such as mainly anaphylaxis and inflammation, but also asthma, acute respiratory distress syndrome, respiratory infections, bronchitis, cardiovascular disease, myocardial infarction, arthritis, pain, shock, sepsis, autoimmune diseases, multiple organ dysfunction syndrome, inflammatory bowel disease, ischemia, ophthalmic inflammation, psoriasis, neoplastic diseases, ulcerative colitis, allergies, Crohn’s disease, cystic fibrosis, emphy Sema, or gastrointestinal ulceration. In addition, PAF mediates platelet aggregation, while PAF antagonists protect amyloid-beta damaged neurons from microglia mediated death [32]. Interestingly, DKPs can play a role of PAF inhibitors useful in the treatment of all these dysfunctions [32,294]. More specifically, cyclo(His-Pro), the same as cyclo(Pro-Gly), cyclo(Pro-Ala), cyclo(Pro-Ile), cyclo(Pro-Leu), cyclo(Pro-Ser), cyclo(Pro-Glu), cyclo(Pro-Gln), cyclo(Pro-Cys), cyclo(Pro-Met), cyclo(Pro-Phe), cyclo(Pro-Trp), cyclo(Pro-Tyr), cyclo(Leu-Pro), cyclo(Val-Pro), cyclo(Leu-hydroxy-Pro), and cyclo(Gly-Pro) are a good example of anti-inflammatory agents [74,179,181,295,296,297]. The latter also has anti-nociceptive effect [298], while cyclo(Leu-Pro) has anti-stress, cytoprotective and genoprotective properties, apart from antimicrobial, antifouling, antimutagenic, carcinogenic activities [299]. The majority of mentioned proline-based DKPs, are isolated from *Pseudomonas* sp., and inhibit pro-inflammatory cytokines, alleviating crystal-induced renal injury. In other words, they possess protective effect in renal nephropathy. With this background, they are non-cytotoxic, and promising in the development of drugs to treat diverse chronic inflammatory diseases, with cyclo(Val-Pro) as a new lead [297]. Furthermore, cyclo(Pro-Tyr) is an example of promising anti-platelet drugs against influenza A [300,301]. On the other hand, DKP alkaloids from *Aspergillus* sp., aspechinulins, have antiviral activity and either an anti-inflammatory or antibacterial effect [178,302]. 

In the case of asthma, we should mention theophylline, a natural substance also used in the treatment of other respiratory disorders, such as chronic obstructive pulmonary disease [303].

Cyclo(Pro-Ile), cyclo(Pro-Leu), cyclo(Pro-Trp) exhibited inhibition of 12-O-tetradecanoylphorbol-13-acetate (TPA) inducing ear edema. They can be useful in the treatment of inflammatory disorders such as arthritis, rheumatism, myalgia, and allergic dermatitis [21].

#### 4.2.11. Cardio-Metabolic Disorders

There are several reports in the literature describing the relevance of proline-based DKPs in cardio-metabolic dysfunctions. Firstly, the calcium-channel-blocking effect of DKPs was proposed [304] with cyclo(Trp-Pro) blocking channels, and confirmed later [192]. In addition, verrucologens from fungus *Penicillium verruculosum* are neurotoxins that can block Ca^2+^ activated K^+^ channels [305]. The ability of blocking sodium channels by DKPs should not be neglected [306]. Next, the potential of DKPs in the treatment of cardiological dysfunctions with an antiarrhythmic effect of cyclo(Trp-Pro) was presented [307]. They have relevance in the treatment of hypertension, and congestive heart failure [306]. Cyclic glycine-proline, a natural nutrient, normalizes blood pressure in obese rats [186]. Cyclo(l-His-l-Pro), mentioned earlier, also plays a cardiovascular role. It exhibits endocrine and electrophysiological effects [176]. Notably, ~430 million people worldwide suffer from diabetes, and it will reach the seventh cause of death by 2030 [308,309]. Current drugs are not sufficient because of the side effects, such as weight gain, liver damage, and allergic reactions [309]. Antihyperglycaemic activity of DKPs is well-documented in the literature [310,311]. Proline-based DKP alkaloids, from marine fungi, can provide a new platform for the discovery of a drug [312]. Cyclo(His-Pro) decreases the level of blood glucose [176].

Pancreatic lipase is an important enzyme in relation to the digestion of dietary triglycerides. Thus, its inhibition is a promising target in the context of obesity treatment. Cyclo(d-8-acetoxyl-Pro-l-Leu), isolated from the *Streptomyces* sp. revealed preferable anti-lipase activity, and could be well bound with the catalytic pocket of the pancreatic lipase [14,313].

Interestingly, cyclo(l-His-l-Pro) inhibits mainly appetite but also the desire for alcohol [21,314].

#### 4.2.12. Hemo-Regulation

*N*-subsituted DKPs dimeric forms, including linker of di-proline-based DKP entities and 1,4-phenylenedimethyl, show hemo-regulatory activities, and may be useful to stimulate hematopoiesis [21,315].

#### 4.2.13. FSH Receptor Antagonists

Among the glycoprotein hormones used in infertility treatment, follicle-stimulating hormone (FSH) is the major value driver for therapeutic intervention. Without the FSH treatment, there is no ovarian hyperstimulation. DKPs can modulate FSH receptor and were considered as lead structures [8,316,317].

#### 4.2.14. Immunosupressants

Some DKPs possess immunosuppressive properties. In particular, chetomin and gliotoxin are of relevance to the inhibition of macrophage phagocytosis, mitogen-activated T cell proliferation [283,318]. Other known examples are tardioxopiperazines, from the *Ascomycete*, and cristatin A, from the plant *Lepidagathis cristata* [32].

#### 4.2.15. Heat Shock Response

Asparaprolines, asparagus-derived proline-containing 3-alkyldiketopiperazines, mainly cyclo(l-Phe-l-Pro), cyclo(l-Tyr-l-Pro), and cyclo(l-Leu-l-Pro), exhibit heat shock protein 70 mRNA induction activity. In more details, the heat shock response, a highly conserved mechanism in organisms, play important role in resetting of the circadian clock in relation to extreme proteotoxic insults (heat, oxidative stress, ultraviolet radiation, heavy metals, toxins, and bacterial infection). The response is mediated at the transcriptional level via heat shock elements, in the upstream area of genes encoding heat shock proteins. Chaperons, members of heat shock protein 70, cause cytoprotective activity through increased cell viability and promotion of protein damage repair. Thus, thanks to antiapoptotic and anti-inflammatory properties, heat shock proteins are therapeutic targets in relation to inter alia regulation of circadian rhythm and sleep, antiulcer drugs, prevention/treatment of diseases caused by protein misfolding such as neurodegenerative disorders or cystic fibrosis [319].

#### 4.2.16. Photoprotective Agents

DKP derivatives have skin photoprotective potential as safe and more eco-friendly agents [320].

#### 4.2.17. Activators of Dioxygen in Oxidative Processes

Proline-based DKPs, including aromaric derivatives, act as catalytic mediators for chemoselective aerobic oxidation processes, such as sulfides oxidation, alkenes epoxidation, or the oxidative coupling of phenols [321]. Cyclo(Pro-Pro) is an archaic precursor in the evolution of life, rendering its potential role in the activation of dioxygen of tremendous importance. Proline ‘could condense to afford DKPs under potentially prebiotic alkaline aqueous conditions with high yield’ [5,38].

#### 4.2.18. Biological Herbicides and Plant Growth Regulation

Synthetic herbicides can lead to increased human health risks, environmental pollution, and the evolution of resistant weeds. Thus, bio-herbicides from natural sources are required. DKPs are promising nature-inspired herbicides due to their environmental friendliness, safety, high selectivity, and probable new target sites [322,323,324]. Proline-derived DKPs have specific herbicide activities [32]. Notably, proline is linked to diverse plant stresses as defense against toxicity [325]. It is used by a plethora of organisms in protection against the cellular imbalance resulting from environmental stress [326]. The proline improves the formation of reactive oxygen species, signaling, cellular apoptosis [327,328,329]. The maculosin-like peptides cyclo(l-Tyr-l-Pro), but also cyclo(l-Phe-l-Pro) produced by *Alternaria alternata* were investigated as potent safe, and environmentally friendly agents [32,91]. Thus, proline-based maculosin and its analogs are a specific herbicide [21]. 

DKPs containing acylhydrazone exhibit insecticidal activity to *Plutella xylostella* and *Culex pipiens pallens* [330]. DKP alkaloids are important secondary metabolites of microbes. Indole DKP alkaloids are the condensation products of tryptophan with a second amino acid-like l-proline [105,312,331].

Okaramines can play the role of insecticide [32]. 

DKPs within a 4-hydroxyproline, such as cyclo(l-Phe-l-4R-Hyp), cyclo(l-Leu-l-4R-Hyp) and cyclo(l-Ala-l-4R-Hyp), isolated from *Alternaria alternata*, are active against the pathogen fungus *Plasmopara viticola*, but non-toxic for the vine plant [21].

Cyclo(d-Pro-l-Val), cyclo(l-Pro-l-Ile), cyclo(l-Pro-l-Phe) are useful in the treatment of tree diseases, which devastate pine forests worldwide [28].

The hydroxyproline analog cyclo(l-Pro-l-trans-OH-Pro), isolated from a *Ruegeria* strain of bacteria, acts towards plant-growth regulation [32]. Furthermore, brevicompanine, proline containing DKP-derived compound from fungus *Penicillium brevicompactum.* plays a role of plant growth promoter [332].

#### 4.2.19. Biotechnological Applications

##### Bone Tissue Engineering

It is a rapidly growing field with the objective of generating a bio-functional tissue, which is able to treat diseased or damaged tissues. The number of orthopedic surgery procedures is projected to ~28 million by 2022 [333]. Therefore, alternative scaffold fabrication techniques are extremely needed in organizing the final, better structure-mechanical properties, and biological response of the implanted biomaterials [334]. DKPs are potential scaffolds for bone regeneration. They can be used in various bone-related applications, inter alia bone defect, and fracture healing, osteoporosis, osteopenia, and many more [333]. DKPs induce osteogenic differentiation, and support osteogenesis with no cytotoxicity. DKP-like compounds (e.g., protuboxepins) are promising inhibitors of induced osteoblastic differentiation. They can offer benefits for fibrodysplasia ossificans progressiva. It is a rare disease causing progressive and widespread heterotopic ossification in soft tissues (muscle, tendons, and ligaments), but surgery is prohibited due to severe heterotopic ossification induced by injury to soft tissues. On the other hand, no drugs have been approved so far. Therefore, DKP-derived agents give a glimmer of hope [335].

##### Vaccinology

Vaccines are among most successful medical interventions. Vaccines recruit the body’s immune system to protect the host from infections or cancer. Modern vaccine formulations are typically comprised of antigen (immunogen) and adjuvant (immune stimulating component) [336]. The antigens are fragments of the targeted pathogen against which an immune system builds the response. Peptide epitopes are minimal antigens and can be divided into two groups: T-cell epitopes, which trigger either cellular or assisting/helper immunity, and B-cell epitopes, which induce humoral immunity [337,338]. Protein antigens typically include several of these epitopes [337,339]. Both peptides and proteins are often inadequately immunogenic, thus adjuvant are required to boost the immunogenicity of vaccine antigen(s) [340]. A wide variety of molecules have been investigated as potential adjuvants for vaccine delivery, including peptides [341,342,343,344,345,346,347,348,349]. 

DKPs have not been used as vaccine antigens to date. However, they are structurally rigid and easy of modification, thus, could act as peptidomimetic antigens in the future [55,350]. B-Cell epitopes are predominantly discontinuous, and the quality of the antibody response depends on the antigen structural conformation. Therefore, the conformational properties and customization ability of DKP might be employed for antigen design [11,351]. DKP can facilitates orienting amino acid’s side chains at ‘stereo-correct’ distances and angles to superimpose those of native epitopes. Such antigens could additionally be tailored to resist enzymatic degradation [352], thus adapted for oral delivery, and can bear additional targeting moieties for improved delivery to lymph nodes. Further, the rigid structures of DKP-based peptidomimetic antigens could improve immune recognition of epitope’s parent protein compared to the highly flexible standard peptide antigens. 

In the vaccine design DKP can be employed not only as peptidomimetic antigens. For example, DKPs have been recently used as transfection agents in mRNA-based vaccine formulations [353]. The formulation was based on a phospholipid-based lipid nanoparticles (LNPs) loaded with mRNA that encoded a pathogenic antigen, intended to transfect host cells. Upon immunization with such vaccine, host cell should express the antigen, which is then recognized by immune cells initiating the immune response. To achieve it, the vaccine formulation must escape endosomes, where mRNA could degrade if exposed to enzymes. Therefore, charge altering DKPs were used as endosome escape agents in this mRNA-LNP formulation to generate proton sponge effect enabling translation of mRNA cargo [353]. The DKP-bearing vaccine was highly immunogenic and triggered a long-lasting Th1-responses, with increased IFN-γ^+^ CD4^+^ and CD8^+^ T-cell counts and generated high IgG titers.

Natively derived DKPs were recently employed as immunomodulatory agents; to alleviate adverse events or disorders arising from inflammatory immune responses. Aspartyl-alanyl DKP fragment in human serum albumin was found to trigger immunomodulatory effect, via RIP-1-dependent suppression of IFN-γ expression from T-cells, rendering them anergic [354]. A self-administrable, anti-inflammatory intranasal product based on this DKP fragment was launched (Ampion^®^) [355], and currently being investigated for alleviating of severe symptoms in COVID-19 patients (Phase-1, NCT04839965). 

##### DNA Delivery

DKP-based cell penetrating peptidomimetics enable efficient cellular uptake and DNA delivery, via specific noncovalent interactions [74,356]. Incorporation of cyclic peptides improves tolerability against proteases [45]. These mimetics contain lysine and aspartic acid [356]. However, proline-containing cell-penetrating peptides present the greatest penetrating activity, among amino acids, in relation to effective transport of DNA [357,358].

##### Fluorescent DKPs

They are excellent future scaffolds to form optical agents useful in inter alia smart molecular bio-imaging or activatable fluorescent pro-drugs, while other applications will appear in the coming years. More specifically, drug discovery programs for cell reprogramming or mechanistic studies to characterize cells in tissue microenvironments and clinical applications are the main areas where highly specific probes are needed. They can be helpful in building dual probes with a multimodal character, compatible with diverse imaging modalities, as well as probes of theranostics to deliver imaging reporters, or therapeutic loads. Fluorescent DKPs will play an important role in the development of new smart therapies for personalized medicine. Moreover, proline-containing DKPs, providing extra bio- and conformational features, have potential. We can mention nano-chromophores on based on the self-assembled proline-based cyclo-dipeptides or fluorophore-bearing oligomers of DKPs for the intracellular DNA delivery [359]. Notably, the self-assembled, well-ordered structure of DKPs leads to much better fluorescent properties [360].

##### Inhalable Powder Formulations

DKPs are utilized in the development of innovative forms of delivery systems, such as technosphere dry powder formulation for drugs administration into the pulmonary system. The formulation of insulin-loaded fumaryl DKP microparticles suitable for inhaled delivery (Afrezza) is a good example [74,361]. It is an interesting, new option for subcutaneous injection, which is an inconvenience for patients, and lead to many side effects such as fat atrophy or fat hyperplasia at the injection site, and unsatisfactory blood glucose control [362]. In comparison with other non-injection routes (oral preparation, transdermal patch, pulmonary inhalation, or nasal mucosal administration) it is better due to its physiological advantage. More specifically, alveoli have a large surface area, high permeability, low toxicity, and large circulation perfusion, good lung deposition inhalation, which could rapidly decrease the blood glucose level without immune stimulation, and the drug can be quickly absorbed into the blood circulation, preventing both drug decomposition by digestive intestine enzyme and ‘first pass effect’ of the liver [74,361,362].

##### Pheromonal Scaffold

Diatoms, an important ecological group of phytoplankton, an extremely diverse group of microalgae, offer possibilities for biotechnological applications, e.g., sustainable pest control in agriculture [363,364]. The proline derived DKP, cyclo(l-Pro-l-Pro), has been recently reported as the first molecular structure of sex-inducing pheromone from marine diatom, *Seminavis robusta* sp., with moderate to good bioactivity, and low to no phytotoxicity [365]. Interestingly, diatom-based DKPs have relevance in biofuel production [366].

## 5. Supramolecular Structuring of DKPs

Cyclo-dipeptides, as important biomolecules, provide promising minimalistic scaffolds for self-assemblies in terms of molecular recognition, aqueous processability, thermal stability, biocompatibility, structural, and functional versatility towards developing innovative therapeutic modalities, with increased efficacy and reduced side effects. A self-assembly is a process of linking molecules via noncovalent (supramolecular) interactions into (bio)functional systems (assemblies, called supermolecules). It can be observed in living organisms, from bacteria to humans. It is useful in arranging bio-machinery, including the DNA double helix, ribosomes, the quaternary structure of enzymes as well as bio-entities, such as cell membranes, the helical structure of collagen, or cytoskeleton [201,367]. Thus, from the supramolecular point of view, non-covalent interactions provide special systems which help to understand bio-processes and create innovative bio-materials that are able to promote 3D cell growth with increased proliferation and differentiation [368]. In this field, DKPs have an inherent tendency to take part in intermolecular interactions driven by specific hydrogen bonding patterns leading to highly organized artificial nanostructures. In particular, DKPs contain four main H-bonding sites, well-oriented H-bond donors, and acceptors, enabling the formation of gelators. In addition, weak forces, such as π⋯π stacking, hydrophobic effect, electrostatic interactions, act as catalysts by templating unique arrangements toward easier gellation [369]. Notably, peptide-based hydrogels are a new great promise for diverse bio-nano-applications, inter alia in tissue engineering or therapeutics delivery [369]. DKP-based supramolecular hydrogelators are promising in anti-infectious vaccination, also in cancer immunotherapy, mainly because they enhance either the vaccines potency or delivery. They can be excellent novel, safer, and more potent adjuvants. In this context, they have high stability, while the costs of production are low. Moreover, they can be produced and modified in large quantities in an easy way [370]. DKP-based low molecular weight gelators also find applications in encapsulation of drugs [74,370]. Self-assembled functional proline containing DKP-based bio-nano-materials have superior structural rigidity and in vivo stability provided by the DKP ring. On the other hand, they display dynamic features such as morphological flexibility [369]. They can cross the blood-brain barrier, and reach difficult pathological targets, offering versatility of desirable biophysical properties. They have broad-spectrum prospectives in bio-medicine, such as high strength biomaterials, isothermal, and injectable gelation in (stimuli-responsive) the delivery of drugs and other bioactive molecules (as well as gene), or 3D bio-printing, smart bio-nano-architectonics, microfluidic devices, bio-sensing, and point-of-care diagnostics, hierarchical biomimetics [367,370]. DKP nanostructures can encapsulate biomolecules, fluorescent dyes and deliver them inside cells. These theranostics have enhanced efficacy [368]. In addition, we should mention proline-rich cyclic peptides which due to size and complexity, can provide useful scaffolds for modulating more challenging biological targets, such as protein–protein interactions and allosteric binding sites [371]. The proline-based DKP scaffold is highly promising and should inspire researchers to develop other appealing smart self-assembled bio-applications toward an effective clinical translation [372]. In particular, targeting domains to promote receptor binding and to achieve cytoplasmic/nuclear engaging drug delivery applications, can be included [368]. Therefore, better understanding of the nature of self-assembly behavior is required [368]. The design of DKP-based self-assembled materials still is a challenge.

From this point of view, a deep insight into the structural landscape of these supramolecular materials can help to understand the modulation mechanism of DKP self-assemblies. The nature of self-assembly depends on the information encoded in the molecular building motifs called synthons [367]. The term ‘molecular synthon’ was introduced by Corey as ‘structural unit within molecules which can be assembled by known or conceivable synthetic operations’ [373], and developed to ‘supramolecular synthon’ by Desiraju as ‘structural units formed with intermolecular interactions’ [374]. Both the self-assembly process and applications of the final supramolecular systems (materials) can be modulated by a suitable recognition and design of synthons [367]. Synthons should be robust enough to exchange from one network to another. The proline-based DKP synthons are a bottom-up approach of great importance, providing inspiring self-assembly building blocks to develop bio-functional architectures with appealing, innovative applications of advanced, next generation theranostics, biomimetics, biomaterials, as well as to tune their features via a proper choice of amino acid and stereochemistry of DKP-synthons [32].

Supramolecular interactions forming building motifs (synthons) are increasingly regarded as useful in the design of ideal ligands, inside the protein pocket [375]. However, they have potential in the design of bio-functional materials, with desired and controllable properties, too.

Supramolecular interactions, despite their importance in bio-systems, have been mostly behind the scenes because of the difficulty of correct detection. Nevertheless, recent advances in structural biology, and better resolution of 3D structures, open a new avenue to deeper insight into supramolecular bio-complexes leading to better knowledge on the bio-supramolecular interactions engaged in the synthons formation. 

In this context, supramoleculas studies on peptides, simple biomolecules, have primary importance. In the course of our ongoing project, focused on the supramolecular perspective of peptide-based systems [6,11,12,52,53,376,377,378,379,380,381,382,383,384,385,386,387,388,389,390,391,392,393,394,395,396,397,398,399,400,401,402,403], a thorough screening of the structural databases revealed an appealing sub-family of proline-based DKPs and its analogs. Here, we provide a brief overview of structural databases and library of family of proline-based DKP structures as well as key non-covalent interaction motifs in found compounds, included in the Appendix A. Notably, the same motifs are observed in DKP-based bio-complexes (Figure 2). These findings can be helpful in the design of both more effective drugs and smart supramolecular bio-materials. 

## 6. Conclusions

To sum up, cyclic dipeptides offer appealing structural and biological diversity. Most recently, more and more interesting DKPs and their derivatives have been isolated from natural sources and investigated in relation to novel, impressive bio-functionalities. The proline-based DKPs are valuable molecular and supramolecular scaffolds, ‘programmed’ by nature, in synthetic biology and protein engineering, toward tuning either the desirable features of modern theranostics, biomimetics, biomaterials, or interactions via a proper choice of additional amino acid and stereochemistry of DKP-synthons, which may be further chemically modified to increase their bio-activity spectrum. Here, we summarize an overview on both the bio-landscape and supramolecular structuring of proline-based DKPs and their derivatives on the basis of the latest scientific and patent literature as well as structural databases. Proline-based cyclo-dipeptides are extremely compact and stable bio-molecules, making noncovalent self-assemblies extra rigid, biomimetic and smart materials to address complex biological problems, towards innovative biomedical and biotechnological applications for personalized medicine. Taken all together, we hope that this guide will deepen the knowledge on proline-containing DKPs and catalyse further intriguing studies in the field of drug discovery utilizing proline-DKP as a versatile scaffold for the molecular and supramolecular design of innovative smart therapeutics and bio-functional self-assembled (nano)materials that have been challenging so far.

## Figures and Tables

**Figure 1 biomolecules-11-01515-f001:**
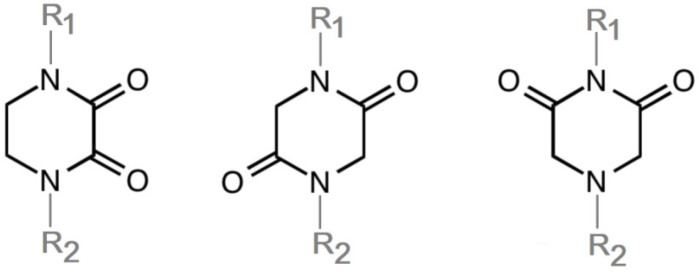
Structure of 2,3-DKP (**left**), 2,5-DKP (**middle**) and 2,6-DKP (**right**) rings as important pharmacophores.

**Figure 2 biomolecules-11-01515-f002:**
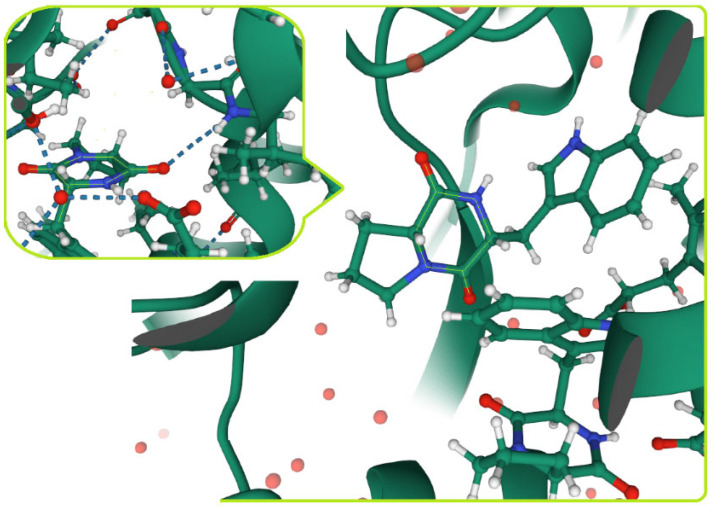
Crystal structure of cytochrome P450 NasF5053 Q65I-A86G mutant variant from *Streptomyces* sp. NRRL F-5053 in the cyclo(l-Trp-l-Pro)-bound state; RCSB PDB ref. code: 6VZA.pdb [234]. On the left: The molecular view of supramolecular interactions, showing, e.g., synthon formed by _(DKP)_C = O H-N H-bonding interaction.

**Table 1 biomolecules-11-01515-t001:** Bi- and tricyclic proline-based DKP with simple side chains.

Entry	Structure	Name	Cell Line	Activity IC_50_ [μM]	Reference
1.	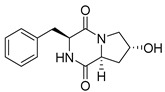 **1**	cyclo(l-Phe-l-Hyp)	U87-MG	5.8 ± 1.7	[105]
U251	18.6 ± 0.1	[105]
HCT-116	>25 ^1^	[106]
OVCAR-8	>25 ^1^	[106]
SF-295	>25 ^1^	[106]
2.	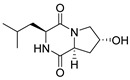 **2**	cyclo(l-Leu-l-Hyp)	U87-MG	14.5 ± 1.6	[105]
U251	29.4 ± 1.3	[105]
3.	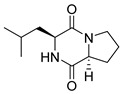 **3**	cyclo(l-Leu-l-Pro)	U87-MG	1.3 ± 0.1	[105]
U251	19.8 ± 0.8	[105]
ECA-109	55 ^2^	[107]
HeLa-S3	41 ^2^	[107]
PANC-1	14 ^2^	[107]
HCT-116	16 ^1^	[108]
HepG2	≥50 ^1^	[108]
MCF7	30 ^1^	[108]
4.	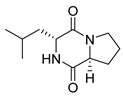 **4**	cyclo(d-Leu-l-Pro)	ECA-109	44 ^2^	[107]
HeLa-S3	52 ^2^	[107]
PANC-1	55 ^2^	[107]
5.	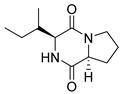 **5**	cyclo(l-Ile-l-Pro)	ECA-109	50 ^2^	[107]
HeLa-S3	45 ^2^	[107]
PANC-1	56 ^2^	[107]
HCT-116	22 ^1^	[108]
HepG2	≥50 ^1^	[108]
MCF7	27 ^1^	[108]
6.	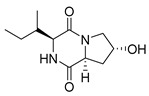 **6**	cyclo(l-Ile-l-Hyp)	ECA-109	54 ^2^	[107]
HeLa-S3	47 ^2^	[107]
PANC-1	42 ^2^	[107]
7.	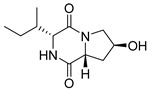 **7**	cyclo(4-*S*-hydroxy-d-Pro-d-Ile)	SF-268	>295 ^3^	[109]
MCF-7	204 ^3^	[109]
H460	234 ^3^	[109]
HT-29	270 ^3^	[109]
CHO-K1	>295 ^3^	[109]
8.	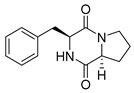 **8**	cyclo(l-Phe-l-Pro)	HCT-116	21.4 ^1^	[106]
OVCAR-8	18.3 ^1^	[106]
SF-295	16.0 ^1^	[106]
ECA-109	42 ^2^	[107]
HeLa-S3	36 ^2^	[107]
PANC-1	50 ^2^	[107]
9.	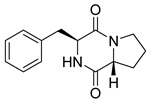 **9**	cyclo(l-Phe-d-Pro)	HCT-116	38.9	[110]
HepG2	≥50	[110]
MCF-7	102.0	[110]
10.	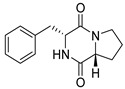 10	cyclo(d-Phe-d-Pro)	HCT-116	94.0	[110]
HepG2	≥50	[110]
MCF-7	114.0	[110]
11.	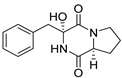 **11**	Penicillatide B	HCT-116	23.0	[110]
HepG2	≥50	[110]
MCF-7	≥50	[110]
12.	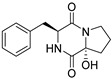 **12**	cyclo(l-Phe-2OH-d-Pro)	HCT-116	30 ^1^	[108]
HepG2	≥50 ^1^	[108]
MCF7	30 ^1^	[108]
13.	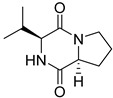 **13**	cyclo(l-Val-l-Pro)	HeLa	33.3 ^4^	[111]
K562	n.a.	[111]
HL-60	n.a.	[111]
BGC-823	n.a.	[111]
MCF-7	n.a.	[111]
14.	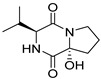 **14**	Bacillusamide B	HCT-116	25 ^1^	[108]
HepG2	≥50 ^1^	[108]
MCF7	27 ^1^	[108]
15.	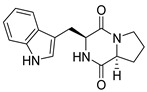 **15**	Brevianamide F, cyclo(l-Trp-l-Pro)	HCT-116	>25 ^1^	[106]
OVCAR-8	11.9	[106]
SF-295	>25 ^1^	[106]
16.	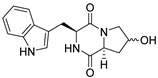 **16**	cyclo(l-Trp-l-Hyp)	HL-60	64.34	[112]
17.	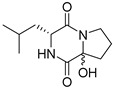 **17**	cyclo(d-Leu-2-OH-Pro)	HL-60	98.49	[112]
18.	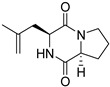 **18**	Penicimutide	HeLa	39.4 ^5^	[111]
K562	n.a.	[111]
HL-60	n.a.	[111]
BGC-823	n.a.	[111]
MCF-7	n.a.	[111]
19.	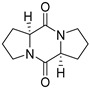 **19**	cyclo(l-Pro-l-Pro)	ECA-109	30 ^2^	[107]
HeLa-S3	40 ^2^	[107]
PANC-1	34 ^2^	[107]

^1^ IC_50_ [μg/mL]; ^2^ the inhibitory effect at 20 μM, inhibition rate (%); ^3^ GI_50_ (μM); ^4^ inhbition rate [%] at at 100 μg/mL (510.2 μM); ^5^ IR % at at 100 μg/mL (480.8 μM); n.a.—not active.

**Table 2 biomolecules-11-01515-t002:** Bicyclic proline-based DKP modified with indole-based side chains.

Entry	Structure	Name	Cell Line	Cytotoxic Effect	Reference
1	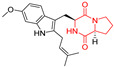 **21**	Tryprostatin A(Try A)	H520	80.1 ± 4.1 ^1^, 79.4 ± 4.2 ^2^	[114]
MCF7	>100 ^1^, 95.0 ± 4.7 ^2^	[114]
PC-3	99.2 ± 4.2 ^1^, 95.6 ± 5.0 ^2^	[114]
2	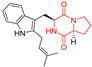 **22**	Tryprostatin B(Try B)	H520	77.6 ± 3.6 ^1^, 60.5 ± 3.5 ^2^	[114]
MCF7	88.2 ± 5.8 ^1^, 66.7 ± 5.3 ^2^	[114]
PC-3	95.5 ± 2.8 ^1^, 68.9 ± 6.6 ^2^	[114]
3	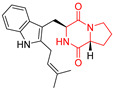 **23**	ds2-TryB	H520	88.3 ± 8.4 ^1^, 0.1 ± 0.1 ^2^	[114]
MCF7	73.6 ± 5.3 ^1^, 0.0 ± 0.0 ^2^	[114]
PC-3	59.3 ± 3.9 ^1^, 0.2 ± 0.0 ^2^	[114]
H520	11.9 ^3^	[115]
MCF7	17.0 ^3^	[115]
PC-3	12.3 ^3^	[115]
4	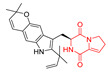 **24**	Piscarinin A	L929	>50 ^4^	[116]
HeLa	>50 ^4^	[116]
LNCAP	2.195 ^5^	[117]
5	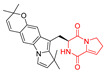 **25**	Piscarinin B	L929	>50 ^4^	[116]
HeLa	>50 ^4^	[116]
LNCAP	1.914 ^5^	[117]
6	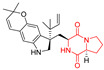 **26**	Notoamide C	HeLa	50 ^5^	[118]
L1210	22 ^5^	[118]
7	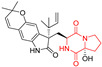 **27**	Notoamide M	22Rv1	55 ^6^	[119]
8	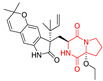 **28**	17-*O*-ethylnotoamide M	22Rv1	25 ^6^	[119]
9	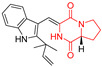 **29**	Brevianamide W	P388	n.a.	[120]
BEL-7402	n.a.	[120]
MOLT	n.a.	[120]
10	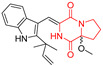 **30**	Brevianamide Q	P388	n.a.	[120]
BEL-7402	n.a.	[120]
MOLT	n.a.	[120]
11	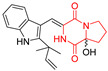 **31**	Brevianamide R	P388	n.a.	[120]
BEL-7402	n.a.	[120]
MOLT	n.a.	[120]
12	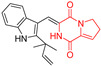 **32**	Brevianamide K	P388	n.a.	[120]
BEL-7402	n.a.	[120]
MOLT	n.a.	[120]
13	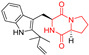 **33**	Brevianamide E	P388	n.a.	[120]
BEL-7402	n.a.	[120]
MOLT	n.a.	[120]

^1^ cell growth inhibition at 10 μM, percent cell survival; ^2^ cell growth inhibition at 100 μM, percent cell survival; ^3^ growth inhibition (GI_50_) in μM; ^4^ IC_50_ [mg/mL]; ^5^ IC_50_ [μg/mL]; ^6^ decreased colony formation at concentrations of 10 μM; n.a.—not active.

**Table 3 biomolecules-11-01515-t003:** Tetracyclic proline-based DKP.

Entry	Structure	Name	Cell Line	IC_50_ [μM]	Reference
**Tetracyclic**					
1.	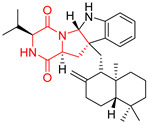 **34**	Drimentine G	A2780	2.81 ± 0.09	[121]
				1.38 ± 0.27	[121]
				>10	[121]
				1.01 ± 0.04	[121]
				2.54 ± 0.18	[121]
2.	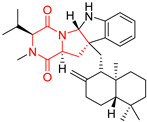 **35**	Drimentine F	BGC-823	>10	[121]
			A549	>10	[121]
			A2780	>10	[121]
3.	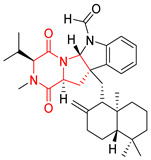 **36**	Drimentine H	Bel-7402	>10	[122]
			BGC-823	>10	[122]
			A549	>10	[122]
			A2780	>10	[122]
4.	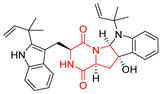 **37**	Okaramine C	L5178Y	14.7	[123]
5.	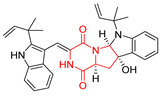 **38**	Okaramine G	L5178Y	12.8	[123]
6.	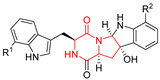 **39**–**41**	Okaramine S (**39**) (R^1^ = R^2^ = prenyl) Okaramine T (**40**) (R^1^ = prenyl, R^2^ = H) Okaramine U (**41**) (R^1^ = R^2^ = H)	(for **39**)	0.78 22.4	[124]
7	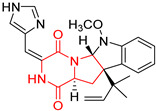 **42**	Roquefortine F	A-549	14.0	[125]
			HL-60	33.6	[125]
			BEL-7402	13.0	[125]
			MOLT-4	21.2	[125]
8.	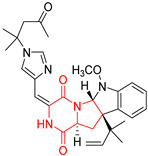 **43**	Roquefortine G	A-549	42.5	[125]
			HL-60	36.6	[125]
			BEL-7402	>50	[125]
			MOLT-4	>50	[125]
9.	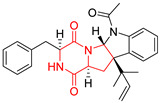 **44**	Fructigenine A	MCF-7	>100, 63.1 ^1^	[126]
			K562	>100, 40.2 ^1^	[126]
			HL-60	>100, 47.7 ^1^	[126]
10.	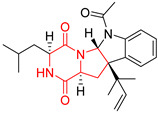 **45**	Fructigenine B	HeLa	>100, 26.6 ^1^	[126]
			BGC-823	>100, 35.4 ^1^	[126]
			MCF-7	>100, 39.6 ^1^	[126]
			K562	>100,49.9 ^1^	[126]
			HL-60	>100, 34.2 ^1^	[126]
11.	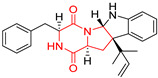 **46**	Rugulosuvine A	HeLa	>100, 52.2 ^1^	[126]
			BGC-823	>100, 52.9 ^1^	[126]
			MCF-7	>100, 62.2 ^1^	[126]
			K562	>100, 75.6 ^1^	[126]
			HL-60	>100, 81.1 ^1^	[126]
12.	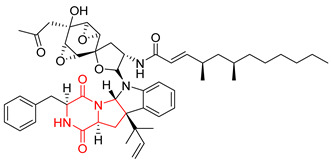 Penicimutanin A (**47**)		BGC-823	8.3, 88.0 ^1^	[126]
			MCF-7	7.3, 84.4 ^1^	[126]
			K562	10.7, 87.4 ^1^	[126]
			HL-60	6.1, 85.2 ^1^	[126]
13.	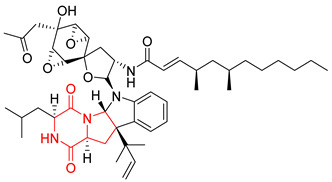 Penicimutanin C (**48**)		HeLa	8.6, 88.1 ^1^	[126]
			BGC-823	8.7, 83.9 ^1^	[126]
			MCF-7	6.0, 80.5 ^1^	[126]
			K562	11.9, 87.7 ^1^	[126]
			HL-60	5.0, 87.3 ^1^	[126]
14.	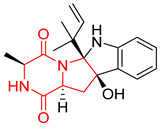 **49**	Eurotiumin A	SF-268	>100	[127]
			HepG2	>100	[127]
15.	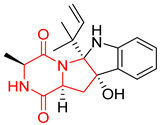 **50**	Eurotiumin B	SF-268	>100	[127]
			HepG2	>100	[127]

^1^ IR% values at 100 μg/mL.

**Table 4 biomolecules-11-01515-t004:** Penta- and hexacyclic proline-based DKP.

Entry	Structure	Name	Cell Line	IC_50_ [μM]	Reference
Pentacyclic					
1.	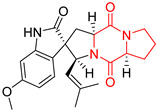 **51**	Spirotryprostatin A	tsFT210	197.5	[128]
2.	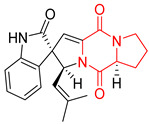 **52**	Spirotryprostatin B	tsFT210	14.0, 12.5 ^1^	[128,129]
			K562	35 ^2^	[129]
			HL-60	10 ^2^	[129]
			3Y1	14.0	[130]
3.	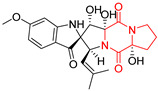 **53**	Spirotryprostatin L	HL-60	6.0	[131]
4.	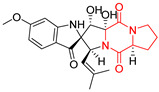 **54**	-	HL-60	7.9	[131]
5.	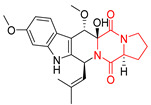 **55**	Cyclotryprostatin B	MCF-7	5.1	[131]
6.	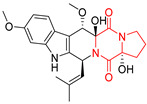 **56**	Cyclotryprostatin F	MCF-7	7.6	[131]
7.	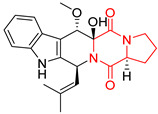 **57**	Cyclotryprostatin G	MCF-7	10.8	[131]
8.	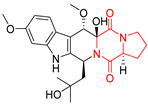 **58**	Cyclotryprostatin E	A549	n.a.	[132]
			A375	n.a.	[132]
			HeLa	n.a.	[132]
9.	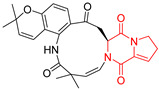 **59**	Versicamide G	HeLa	n.a.	[133]
			HCT-116	n.a.	[133]
			HL-60	n.a.	[133]
			K562	n.a.	[133]
Hexacyclic					
10.	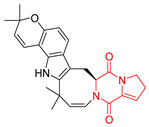 **60**	Versicamide A	HeLa	n.a.	[133]
			HCT-116	n.a.	[133]
			HL-60	n.a.	[133]
			K562	n.a.	[133]
11.	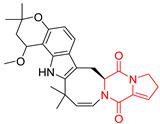 **61**	Versicamide B	HeLa	n.a.	[133]
			HCT-116	n.a.	[133]
			HL-60	n.a.	[133]
			K562	n.a.	[133]
12.	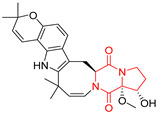 **62**	Versicamide C	HeLa	n.a.	[133]
			HCT-116	n.a.	[133]
			HL-60	n.a.	[133]
			K562	n.a.	[133]
13.	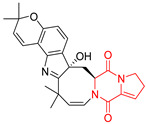 **63**	Versicamide D	HeLa	n.a.	[133]
			HCT-116	n.a.	[133]
			HL-60	n.a.	[133]
			K562	n.a.	[133]
14.	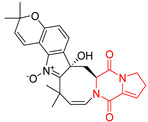 **64**	Versicamide E	HeLa	n.a.	[133]
			HCT-116	n.a.	[133]
			HL-60	n.a.	[133]
			K562	n.a.	[133]
15.	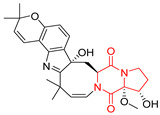 **65**	Versicamide F	HeLa	n.a.	[133]
			HCT-116	n.a.	[133]
			HL-60	n.a.	[133]
			K562	n.a.	[133]
16.	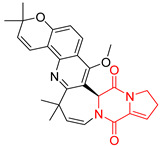 **66**	Versicamide H	HeLa	19.4	[133]
			HCT-116	17.7	[133]
			HL-60	8.7	[133]
			K562	22.4	[133]
17.	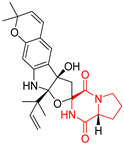 **67**	Asperversiamide I	HeLa	7.3	[134]
18.	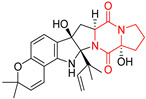 **68**	Speramide B	PC3	>40	[135]
			DU145	>40	[135]
			LNCaP	>40	[135]

^1^ complete inhibition of cell cycle progression [μg/mL]; ^2^ MIC (Minimum Inhibitory Concentration) values for cytotoxicity; n.a.—non active.

**Table 5 biomolecules-11-01515-t005:** Hepta-, polycyclic and dimeric proline-based DKP.

Entry	Structure	Name	Cell Line	IC_50_ [μM]	Reference
Heptacyclic					
1	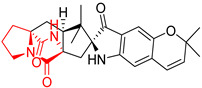 Speramide A (**69**)		PC3	>40	[135]
			DU145	>40	[135]
			LNCaP	>40	[135]
2	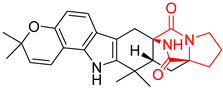 Stephacidin A (**70**)		PC3	2.1	[136]
			LNCaP	1	[136]
			A2780	4	[136]
			A2780/DDP	6.8	[136]
			A2780/Tax	3.6	[136]
			HCT116	2.1	[136]
			HCT116/mdr+	6.7	[136]
			HCT116/topo	13.1	[136]
			MCF-7	4.2	[136]
			SKBR3	2.15	[136]
			LX-1	4.22	[136]
3	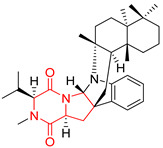 **71**	Drimentine I	HeLa	16.73	[137]
			A549	n.a.	[137]
4	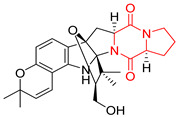 **72**	Gartryprostatin A	MV4-11	7.2	[138]
			K562	> 10, 19.7 ^1^	[138]
			HL-60	> 10, 20.6 ^1^	[138]
			A549	> 10, 15.2 ^1^	[138]
5	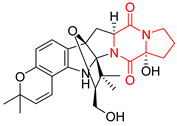 **73**	Gartryprostatin B	MV4-11	10	[138]
			K562	> 10, 18.5 ^1^	[138]
			HL-60	> 10, 23.2 ^1^	[138]
			A549	> 10, 18.2 ^1^	[138]
6	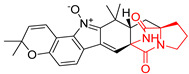 **74**	(+)-Avrainvillamide	HeLa	1.1 ^2^	[87]
			T-47D	0.33 ^3^	[139]
			LNCaP	0.423	[139]
			NB4	1.1	[140]
			HL-60	0.643	[140]
			MV4-11	0.116	[140]
			OCI-AML3	0.112	[140]
			Molm-13	0.078	[140]
7	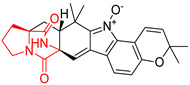 **75**	(−)-Avrainvillamide	T-47D	0.91 ^3^	[139]
			LNCaP	1.4 ^3^	[139]
Polycyclic					
8	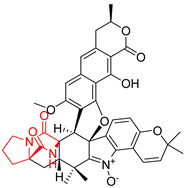 **76**	Waikikiamide A	HT1080	0.519	[141]
			PC3	1.855	[141]
			Jurkat	0.62	[141]
			A2780S	0.78	[141]
Dimeric					
9	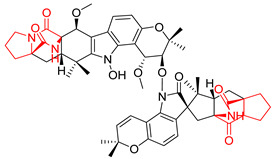 Waikikiamide B (**77**)		HT1080	1.135	[141]
			PC3	1.805	[141]
			Jurkat	1.79	[141]
			A2780S	1.127	[141]
10	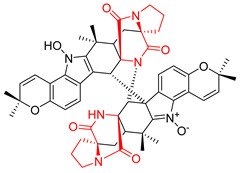 Stephacidin B (**78**)		PC3	0.37	[136]
			LNCaP	0.06	[136]
			A2780	0.33	[136]
			A2780/DDP	0.43	[136]
			A2780/Tax	0.26	[136]
			HCT116	0.46	[136]
			HCT116/mdr+	0.46	[136]
			HCT116/topo	0.42	[136]
			MCF-7	0.27	[136]
			SKBR3	0.32	[136]
			LX-1	0.38	[136]
11	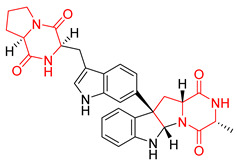 Naseseazine A (**79**)		AGS	n.a.	[142]
			SH-SY5Y	n.a.	[142]
			TF-1	n.a.	[142]
			HT-29	n.a.	[142]
12	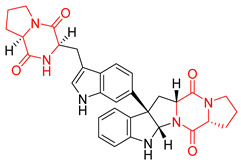 Naseseazine B (**80**)		AGS	n.a.	[142]
			SH-SY5Y	n.a.	[142]
			TF-1	n.a.	[142]
			HT-29	n.a.	[142]
13	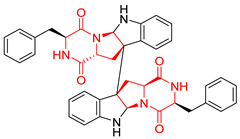 Asperflocin (**81**)		HT-29	>20	[143]
			A375	10.29 ± 2.37	[143]
			MCF-7	>20	[143]
			HepG2	>20	[143]
14	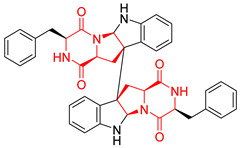 WIN 64821 (**82**)		HT-29	>20	[143]
			A375	>20	[143]
			MCF-7	>20	[143]
			HepG2	>20	[143]

^1^ Inhibition rate at 10 μM; ^2^ IC_90_ [μg/mL]; ^3^ GI_50_ [μM]; n.a.—not active.

**Table 6 biomolecules-11-01515-t006:** Sulfur-containing proline-based DKPs.

Entry	Structure	Name	Cell Line	IC_50_ [μM]	Reference
Tricyclic					
1.	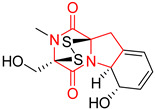 **83**	Gliotoxin	SF-268	0.24 ± 0.10	[144]
			MCF-7	0.08 ± 0.0	[144]
			NCI-H460	0.24 ± 0.01	[144]
			HepG-2	0.21 ± 0.01	[144]
			P388	0.024	[145]
			HEK293	1.58 ± 0.03	[146]
			HCT-116	1.24 ± 0.38	[146]
			RKO	0.80 ± 0.20	[146]
			tsFT210	0.15 ^1^	[147]
			U937	0.20 ± 0.03	[148]
			PC-3	0.39 ± 0.03	[148]
2.	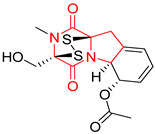 **84**	Acetylgliotoxin	SF-268	0.25 ± 0.03	[144]
			MCF-7	0.22 ± 0.04	[144]
			NCI-H460	0.32 ± 0.02	[144]
			HepG-2	0.49 ± 0.07	[144]
			HEK293	4.49 ± 0.24	[146]
			HCT-116	0.89 ± 0.04	[146]
			RKO	1.24 ± 0.18	[146]
3.	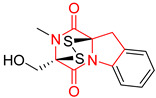 **85**	6-deoxy-5*a*,6-didehydrogliotoxin	SF-268	1.35 ± 0.05	[144]
			MCF-7	0.68 ± 0.02	[144]
			NCI-H460	1.27 ± 0.04	[144]
			HepG-2	1.52 ± 0.03	[144]
			P388	0.058	[145]
4.	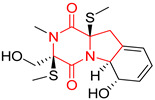 **86**	bisdethiobis(methylthio)gliotoxin	SF-268	>100	[144]
			MCF-7	>100	[144]
			NCI-H460	>100	[144]
			HepG-2	>100	[144]
			P388	0.11	[145]
			HEK293	16.39 ± 0.38	[146]
			HCT-116	8.59 ± 0.96	[146]
			RKO	10.32 ± 0.04	[146]
			U937	0.52 ± 0.07	[148]
			PC-3	15.87 ± 1.38	[148]
			B16	82 ^2^	[149]
5.	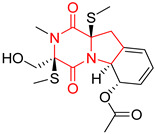 **87**	6-acetylbisdethiobis(methylthio)gliotoxin	SF-268	34.0 ± 3.6	[144]
			MCF-7	3.1 ± 0.10	[144]
			NCI-H460	5.4 ± 0.60	[144]
			HepG-2	7.0 ± 0.17	[144]
			HEK293	>50	[146]
			HCT-116	>50	[146]
			RKO	>50	[146]
6.	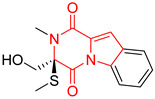 **88**	Dichotocejpin A	SF-268	35.7 ± 2.1	[144]
			MCF-7	29.5 ± 2.3	[144]
			NCI-H460	>100	[144]
			HepG-2	28.9 ± 3.0	[144]
7.	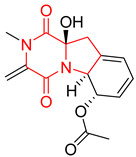 **89**	Dichotocejpin B	SF-268	>100	[144]
			MCF-7	>100	[144]
			NCI-H460	>100	[144]
			HepG-2	>100	[144]
8.	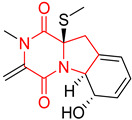 **90**	bis(dethio)-10*a*-methylthio-3*a*-deoxy-3,3*a*-didehydrogliotoxin	P388	3.4	[145]
9.	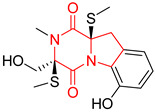 **91**	bis(dethio)bis- (methylthio)-5*a*,6-didehydrogliotoxin	P388	0.11	[145]
			tsFT210	89 ^1^	[147]
			U937	>100	[148]
			PC-3	>100	[148]
			B16	78 ^2^	[149]
10.	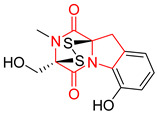 **92**	5*a*,6-didehydrogliotoxin	P388	0.056	[145]
11.	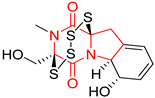 **93**	Gliotoxin G	P388	0.020	[145]
12.	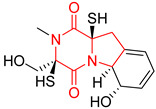 **94**	reduced Gliotoxin	HEK293	1.26 ± 0.04	[146]
			HCT-116	0.43 ± 0.04	[146]
			RKO	0.41 ± 0.07	[146]
13.	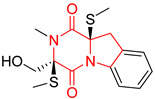 **95**	6-deoxy-bis(dethio)bis- (methylthio)-5*a*,6-didehydrogliotoxin	HEK293	>50	[146]
			HCT-116	>50	[146]
			RKO	>50	[146]
14.	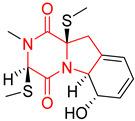 **96**	3-dehydroxymethylbisdethio-3,10*a*-bis(methylthio)gliotoxin	B16	86 ^2^	[149]
15.	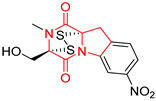 **97**	Glionitrin A	HCT-116	0.82	[150]
			A549	0.55	[150]
			AGS	0.45	[150]
			DU145	0.24	[150]
			MCF-7	2.0	[150]
			HepG2	2.3	[150]
16.	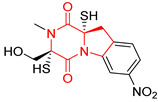 **98**	Glionitrin B	DU145	n.a.	[151]
17.	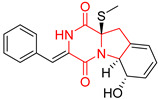 **99**	Phomazine B	HL-60	>10	[152]
			HCT-116	>10	[152]
			A549	>10	[152]
			K562	>10	[152]
			MGC-803	8.5	[152]
tetracyclic					
18.	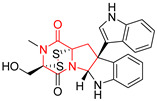 **100**	Luteoalbusin A	SF-268	0.46 ± 0.05	[154]
			MCF-7	0.23 ± 0.03	[154]
			NCI-H460	1.15 ± 0.03	[154]
			HepG-2	0.91 ± 0.03	[154]
19.	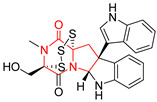 **101**	Luteoalbusin B	SF-268	0.59 ± 0.03	[154]
			MCF-7	0.25 ± 0.00	[154]
			NCI-H460	1.31 ± 0.12	[154]
			HepG-2	1.29 ± 0.16	[154]
20.	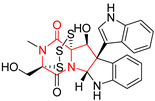 **102**	T988A	SF-268	1.04 ± 0.03	[154]
			MCF-7	0.91 ± 0.03	[154]
			NCI-H460	5.60 ± 0.58	[154]
			HepG-2	3.52 ± 0.74	[154]
21.	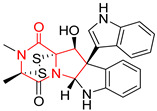 **103**	Gliocladine C	SF-268	0.73 ± 0.05	[154]
			MCF-7	0.23 ± 0.03	[154]
			NCI-H460	6.57 ± 0.81	[154]
			HepG-2	0.53 ± 0.04	[154]
22.	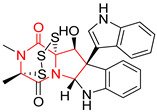 **104**	Gliocladine D	SF-268	2.49 ± 0.07	[154]
			MCF-7	0.65 ± 0.07	[154]
			NCI-H460	17.78 ± 0.27	[154]
			HepG-2	2.03 ± 0.07	[154]
23.	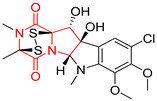 **105**	Sporidesmin A	E006AA-hT	2.5	[155]
pentacyclic					
24.	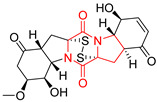 **106**	Brocazine A	Du145	4.2	[156]
			HeLa	6.8	[156]
			HepG2	6.4	[156]
			MCF-7	5.5	[156]
			NCI-H460	4.9	[156]
			SGC-7901	2.6	[156]
			SW1990	6.0	[156]
			SW480	2.0	[156]
			U251	5.2	[156]
25.	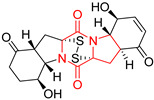 **107**	Brocazine B	Du145	3.6	[156]
			HeLa	5.3	[156]
			HepG2	5.5	[156]
			MCF-7	6.1	[156]
			NCI-H460	4.0	[156]
			SGC-7901	2.4	[156]
			SW1990	6.4	[156]
			SW480	1.2	[156]
			U251	3.5	[156]
26.	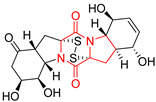 **108**	Brocazine C	Du145	>20	[156]
			HeLa	>20	[156]
			HepG2	>20	[156]
			MCF-7	>20	[156]
			NCI-H460	>20	[156]
			SGC-7901	>20	[156]
			SW1990	>20	[156]
			SW480	>20	[156]
			U251	>20	[156]
27.	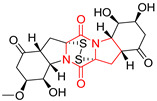 **109**	Brocazine D	Du145	>20	[156]
			HeLa	>20	[156]
			HepG2	>20	[156]
			MCF-7	>20	[156]
			NCI-H460	>20	[156]
			SGC-7901	>20	[156]
			SW1990	>20	[156]
			SW480	>20	[156]
			U251	>20	[156]
28.	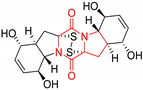 **110**	Brocazine E	Du145	11.2	[156]
			HeLa	4.3	[156]
			HepG2	5.6	[156]
			MCF-7	9.0	[156]
			NCI-H460	12.4	[156]
			SGC-7901	3.3	[156]
			SW1990	2.1	[156]
			SW480	n.t.	[156]
			U251	6.1	[156]
29.	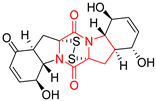 **111**	Brocazine F	Du145	1.7	[156]
			HeLa	6.9	[156]
			HepG2	2.9	[156]
			MCF-7	3.0	[156]
			NCI-H460	0.89	[156]
			SGC-7901	8.0	[156]
			SW1990	5.9	[156]
			SW480	n.t.	[156]
			U251	5.3	[156]
31.	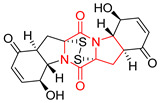 **112**	Epicorazine A	HL-60	0.05	[152]
			HCT-116	0.33	[152]
			A549	2.3	[152]
			K562	1.5	[152]
			MGC-803	2.7	[152]
32	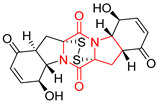 **113**	Epicorazine B	HL-60	1.6	[152]
			HCT-116	>10	[152]
			A549	>10	[152]
			K562	4.6	[152]
			MGC-803	5.2	[152]
33.	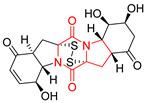 **114**	Epicorazine C	HL-60	3.5	[152]
			HCT-116	7.1	[152]
			A549	>10	[152]
			K562	>10	[152]
			MGC-803	3.4	[152]
34.	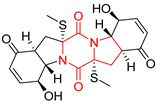 **115**	Exserohilone A	HL-60	3.5	[152]
			HCT-116	4.0	[152]
			A549	>10	[152]
			K562	>10	[152]
			MGC-803	4.2	[152]
35.	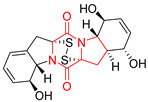 **116**	Brocazine G	A2780	0.664	[157]
			A2780 cisR	0.661	[157]
36.	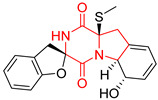 **117**	Spirobrocazine A	A2780	n.a.	[157]
			A2780 cisR	n.a.	[157]
37.	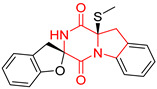 **118**	Spirobrocazine B	A2780	n.a.	[157]
			A2780 cisR	n.a.	[157]
38.	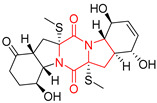 **119**	5′-hydroxy-6′-ene-epicoccin G	Huh.7.5	n.a.	[158]
39.	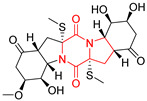 **120**	7-methoxy-7′-hydroxyepicoccin G	Huh.7.5	n.a.	[158]
40.	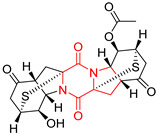 **121**	8′-acetoxyepicoccin D	Huh.7.5	n.a.	[158]
41.	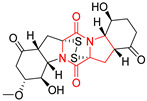 **122**	7′-demethoxyrostratin C	Huh.7.5	9.52	[158]
42.	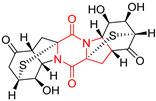 **123**	Epicoccin E	Huh.7.5	n.a.	[158]
43.	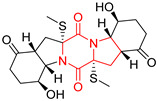 **124**	Epicoccin G	Huh.7.5	n.a.	[158]
44.	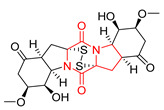 **125**	Rostracin C	Huh.7.5	4.88	[158]
45.	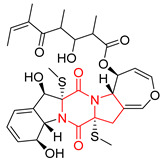 **126**	-	KB	>30.9	[159]
			BC-1	29.2	[159]
			NCI-H187	22.9	[159]
46.	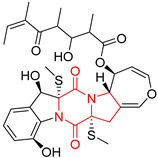 **127**	-	KB	>31.0	[159]
			BC-1	>31.0	[159]
			NCI-H187	20.3	[159]
47.	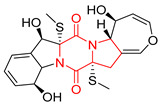 **128**	-	KB	>46.0	[159]
			BC-1	>46.0	[159]
			NCI-H187	>46.0	[159]
dimeric					
48.	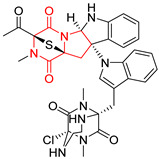 **129**	Cristazine	HeLa	0.5	[160]
			A431	~0.5	[161]
49.	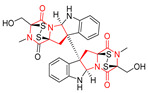 **130**	Chaetocin	HL-60	0.153 ± 0.22	[162]
			U937	0.096 ± 0.18	[162]
			KG-1a	0.103 ± 0.34	[162]
50.	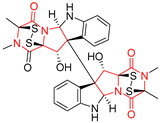 **131**	Verticillin A	HT-29	0.02	[163]
			H460	0.20	[163]
			SF-268	0.25	[163]
			MCF-7	0.37	[163]
			MDAMB-435	0.070, 0.018	[163,164]
			MDA-MB-231	0.023	[164]
			OVCAR3	0.036	[164]
			AGS	0.070	[165]
			HeLa	0.319	[165]
			OVSAHO	0.060	[166]
			OVCAR4	0.047	[166]
			OVCAR8	0.045	[166]
51.	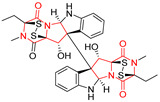 **132**	Verticillin H	HT-29	0.04	[163]
			H460	0.30	[163]
			SF-268	0.33	[163]
			MCF-7	0.49	[163]
			MDAMB-435	0.10, 0.044	[163,164]
			MDA-MB-231	0.031	[164]
			OVCAR3	0.229	[164]
52.	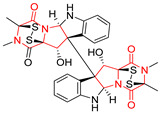 **133**	Sch 52900	HT-29	0.01	[163]
			H460	0.29	[163]
			SF-268	0.37	[163]
			MCF-7	0.58	[163]
			MDAMB-435	0.48	[163]
53.	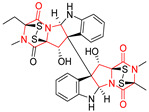 **134**	Sch 52901	HT-29	0.19	[163]
			H460	1.20	[163]
			SF-268	0.75	[163]
			MCF-7	1.11	[163]
			MDAMB-435	0.03	[163]
54.	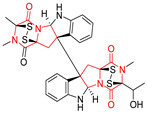 **135**	Gliocladicillin A	HT-29	n.t.	[163]
			H460	0.03	[163]
			SF-268	0.09	[163]
			MCF-7	0.09	[163]
			MDAMB-435	n.t.	[163]
55.	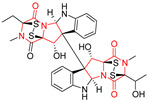 **136**	Gliocladicillin C	HT-29	0.03	[163]
			H460	0.52	[163]
			SF-268	0.38	[163]
			MCF-7	0.61	[163]
			MDAMB-435	0.08	[163]
56.	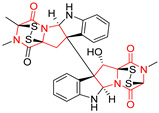 **137**	11′-Deoxyverticillin	HT-29	n.t.	[163]
			H460	0.01	[163]
			SF-268	0.04	[163]
			MCF-7	0.03	[163]
			MDAMB-435	n.t.	[163]
			HCT-116	0.030 ^1^	[167]
57.	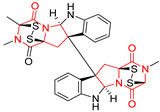 **138**	11,11′-dideoxyverticillin	HCT-116	0.030 ^1^	[167]
			MDAMB-468	0.281 ± 0.022	[168]
			MCF-7	0.158 ± 0.070	[168]
			MDA-MB-435	0.223 ± 0.099	[168]
			MDA-MB-231	0.138 ± 0.025	[168]

^1^ IC_50_ [μg/mL]; ^2^ inhibitory rate at the concentration of 500 μg/mL; n.a.—not active; n.t.—not tested.

**Table 7 biomolecules-11-01515-t007:** Antibacterial DKPs.

Entry	Structure	Name	Bacterial Strains	MIC [μg/mL]	Reference
1.	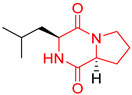	cyclo(l-Leu-l-Pro)	*Enterococcus faecalis (VRE)* *Micrococcus luteus*, *Staphylococcus aureus*	12.5	[214]
				2.9 mmol/L	[218]
2.	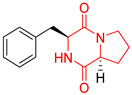	cyclo(l-Phe-l-Pro)	*Micrococcus luteus*, *Staphylococcus aureus* *Staphylococcus aureus*	2.9 mmol/L	[218]
				10 mm zone of inhibition	[222]
3.	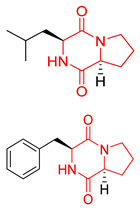	Combination of cyclo(l-Leu-l-Pro) and cyclo(l-Phe-l-Pro)	*Enterococcus faecalis (VRE)* *Micrococcus luteus*, *Escherichia coli, Candida albicans*, *Staphylococcus aureus, Cryptococcus neoformans*	0.25–1	[215]
				0.25–0.5	[215]
4.	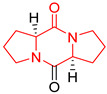	cyclo(l-Pro-l-Pro)	*Pseudomonas aeruginosa*	150 AU/mL	[219]
			*Micrococcus luteus*	100 AU/mL	[219]
5.	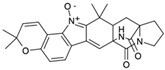	Avrainvillamide	*Staphylococcus aureus*	12.5	[87]
			*Streptococcus pyogenes*	12.5	[87]
			*Enterococcus faecalis*	25	[87]
6.	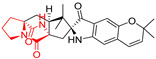	Speramide A	*Pseudomonas aeruginosa*	0.36	[41]
7.	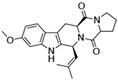	Fumitremorgin C	*Staphylococcus aureus* *Bacillus subtilis*	2.1–3.3	[41]
8.	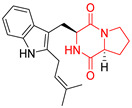	Tryprostatin B	*Staphylococcus aureus* *Bacillus subtilis*	2.1–3.3	[41]
9.	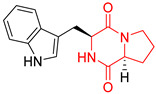	Brevianamide F, cyclo(l-Trp-l-Pro)	*Micrococcus luteus* *Staphylococcus aureus*	Reported as zones of inhibition of 14 mm	[216]
10.	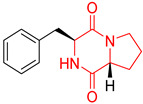	cyclo(l-Phe-d-Pro)	*Vibrio anguillarum*	0.10	[220]
			*Staphylococcus aureus*	14 mm zone of inhibition	[108]
11.	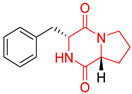	cyclo(d-Phe-d-Pro)	*Vibrio anguillarum*	0.03	[220]

## Data Availability

Not applicable.

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
