# Peer review of "Cyclic Dipeptides: The Biological and Structural Landscape with Special Focus on the Anti-Cancer Proline-Based Scaffold"

_biomolecules, 2021, doi:10.3390/biom11101515_

Round 1

Reviewer 1 Report

Today, a large number of works are devoted to the development of methods for obtaining DKPs, the study of self-assembly of DKPs, and the practical application of materials based on them, as well as the study of the biological activity of DKPs. Such molecules have great potential for medicine, technology, biotechnology, ecology, energy, etc. Chemical modification of DKPs based on proteinogenic amino acids, as well as DKPs based on non-proteinogenic amino acids, significantly expands the horizons of their application.

The reviewers chose proline-based DKPs and their use in anti-cancer treatment. They did a good analysis of the literature and tried to cover all the existing works to date. Moreover, the review also considers other aspects of the application of proline-based DKPs. The text is well structured. Undoubtedly, this review will be of interest to a wide range of readers.

In my opinion, this review can be published after minor modifications.

Here are some minor points for authors.

  1. Lines 57-58 “Proline-containing DKPs are more easily generated compared to other DKPs”. This claim is controversial. First, this statement is given not in the Ref. [13], but in the work [Yoshihisa K. On the mechanism of L-prolyl diketopiperazine formation by streptomyces. Bull Agr Chem Soc Japan. 1960; 24: 449–458.], referred to by the authors of [13]. Moreover, the work dealt only with enzymatic hydrolysis! Currently, there are works on a very simple synthesis of other DKPs using solid-state reactions. For example, works of Smith or Ziganshin. Therefore, this statement must be revised or clarified.
  2. The revision of Table 3 is required. Rows in column 4 have shifted. What can the authors say about the effect of chirality on biological activity? Are there any patterns? It would be interesting to discuss this.
  3. Can the authors make an assumption about the reasons for the loss of biological activity during the transition from compounds 23 to 33 (Table 2)? These compounds have minimal structural differences.
  4. Table 7 needs to be edited. There are duplications and errors in the reference numbers.
  5. Remove the hyphen in DKP titles throughout the text. Instead of “cyclo-(“ should be “cyclo(“
  6. Lines 1203-1204. It is not clear what is the advantage of pulmonary administration vs pulmonary inhalation.

Author Response

Dear Editors,

we sincerely thank you so much for so excellent reviews of our paper. We would like to thank both referees for their opinion and especially the time spent on check our work. It was a truly pleasant surprise for us. We agree with the comments of Referee 2 and we corrected the manuscript according to his suggestions. In particular:

1. We modified mentioned sentence.
2. We revised table 3.

Although it is well known, that the biological activity strongly depends on the
chirality/stereochemistry of the molecule it is difficult to find a general pattern. The source of this phenomenon comes from the situation that bioactive compounds work in the chiral environment of the living organism. The best example concerns the drug Thalidomide, where (R)-Thalidomide is a sedative and analgesic drug, while (S)-Thalidomide is teratogenic and inhibits angiogenesis. Consequently, in our own research (Mieczkowski, A. et al. Bioorganic & Medicinal Chemistry Letters, 2018, 28(4), 618-625) we observed, that only S-enantiomers of investigated Anthramycin analogs exhibit selective antileukemic effect, while R-enantiomers are deprived of cytotoxic activity against leukemia. This issue is very important and must be taken into account when designing new drugs.

3. First of all, compounds 23 and 33 were tested on a completely different panel of cell lines. It’s impossible to compare biological effect in such situation as investigated compounds could exhibit different activity and selectivity when tested on different cell lines. Additionally, the small structural change (like different chirality) could have a great impact on the biological activity of
tested compounds. Compounds 23 and 33 have a different configuration on the chiral center of the bicyclic core as well as different side chain attached to the indole ring. Consequently, these structural changes are sufficient for the compounds to display different biological activity. A similar effect concerns two diastereomeric compounds 22 and 23 differing only in the configuration on
the chiral center of the bicyclic core. Compound 23 exhibits a strong cytotoxic effect, while compound 22 has relatively weak cytotoxicity.

4. Table 7 is corrected.
5. We replaced ‘cyclo-(‘ by ‘cyclo(‘ throughout the text.
6. It was a duplication and mistake, of course. We corrected the sentence.

In addition, we corrected other „glitches” in the paper. All changes are in blue.

Reviewer 2 Report

Bojarska et al  have written excellent cyclic dipeptides, also know as diketopiperazines (DKP).  They have presented an updated review on the biological and structural profile of these appealing biomolecules, with a particular emphasis on those with anticancer properties, since cancers are the main cause of death all over the world. Additionally, they provided a consideration on supramolecular structuring and synthons, based on the proline-based DKP privileged scaffold, for inspiration in the design of compound libraries in search of ideal ligands, innovative self-assembled nanomaterials, and bio- functional architectures.

This is an excellent review for the scientific community and I strongly recommend to accept this review article.

Author Response

Dear Editors,

we sincerely thank you so much for so excellent reviews of our paper. We would like to thank both referees for their opinion and especially the time spent on check our work. It was a truly pleasant surprise for us